# Efficient Allocation of Working Memory Resource for Utility Maximization in Humans and Recurrent Neural Networks

**Qingqing Yang**
Department of Psychology
The Ohio State University
Columbus, OH 43214
yang.6118@osu.edu

**Hsin-Hung Li**
Department of Psychology
The Ohio State University
Columbus, OH 43214
li.14492@osu.edu

## Abstract

Working memory (WM) supports the temporary retention of task-relevant information. It is limited in capacity and inherently noisy. The ability to flexibly allocate WM resource is a hallmark of adaptive behavior. While it is well established that WM resource can be prioritized via selective attention, whether they can be allocated based on reward incentive alone remains under debate—raising open questions about *whether humans can efficiently allocate WM resource based on utility*. To address this, we conducted behavioral experiments using orientations as stimuli. Participants first learned stimulus–reward associations and then performed a delayed estimate WM task. We found that WM precision, indexed by the variability of memory reports, reflected both natural stimulus priors and utility-based allocation. The effects from reward and prior on memory variability both grew over time, indicating their effects in stabilizing memory representations. In contrast, memory bias was largely unaffected by time or reward. To interpret these findings, we extended efficient coding theory by incorporating time and reformulating the objective from minimizing estimation loss to maximizing expected utility. We showed that the behavioral results were consistent with an observer that efficiently allocates WM resource over time to maximize utility. Lastly, we trained recurrent neural networks (RNNs) to perform the same WM task under a 2×2 design: prior (uniform vs. natural) × reward policy (baseline vs. reward context). Human-like behaviors emerged in RNNs: memory was more stable (lower variability) for stimuli associated with higher probability or rewards, and these effects increased over time. Transfer learning showed that recurrent dynamics were crucial for adapting to different priors and reward policies. Together, these results provide converging behavioral and computational evidence that WM resource allocation is shaped by environmental statistics and rewards, offering insight into how intelligent systems can dynamically optimize memory for utility under resource constraints.

## 1 Introduction

Working memory (WM) is the neural and cognitive process that temporarily maintain task-relevant information online[Baddeley, 2003]. It supports a wide range of high-level cognition functions such as learning, decision-making and planning [Collins and Frank, 2012, Daneman and Carpenter, 1980, Süß et al., 2002, Ehrlich and Murray, 2022]. WM has limited capacity and is inherently noisy—the quality of the WM decreases with the number of items stored [Luck and Vogel, 1997, 2013, Ma et al., 2014], and the precision of WM representations decays over time [Pertzov et al., 2017, Panichello

39th Conference on Neural Information Processing Systems (NeurIPS 2025).

et al., 2019, Shin et al., 2017]. Thereby, the ability to flexibly and efficiently allocate WM resource is essential for adaptive and goal-directed behaviors.

It is well-established that humans can allocate WM resource in response to attentional cues. When multiple items must be remembered, people can prioritize certain items over others, suggesting a degree of control over the distribution of WM resource [Zhang and Luck, 2008, Emrich et al., 2017, Dube et al., 2017, Klyszejko et al., 2014, Yoo et al., 2018, Bays, 2014]. However, a more fundamental and still unresolved question is whether WM resource is allocated efficiently to maximize reward or utility—the common currency of the subjective value. Recent studies have reported mixed results regarding whether WM resource can be modulated based on reward alone [Brissenden et al., 2023, Van den Berg et al., 2023, Weiss et al., 2025, Manga et al., 2020], when factors such as spatial attention are controlled. These findings cast doubt on the notion that WM resource can be flexibly optimized to maximize an individual's utility. In parallel, although both cognitive models [Van den Berg et al., 2014, Bays et al., 2024] and recurrent neural network (RNN) models have been developed to account for various aspects of working memory [Wimmer et al., 2014, Compte et al., 2000, Bouchacourt and Buschman, 2019, Esnaola-Acebes et al., 2022, Wang, 2021], it remains largely underexplored how such models should allocate WM resource in order to maximize expected utility, a signature of normative and reward-sensitive computation.

Here, we conducted behavioral experiments to test the hypothesis that after value learning, humans allocate WM resource efficiently according to stimulus-reward association in order to maximize utility. To preview, we found that memory representations were more stable for stimuli associated with higher reward, as well as for those more probable in the natural environment. To explain these findings, we extended efficient coding theory to the time domain and reformulated the objective from minimizing estimation loss to utility maximization, providing a normative account of how WM resource can be allocated over time to optimize expected utility [Wei and Stocker, 2015, Hahn and Wei, 2024, Morais and Pillow, 2018, Schaffner et al., 2023]. Finally, we show that recurrent neural networks (RNNs) trained to maximize reward exhibit human-like behaviors, integrating both prior and reward information in a manner consistent with the behavioral data.

## 2 Joint effects of prior and reward on human working memory

### 2.1 Behavioral tasks

The behavioral experiments consisted of two sessions scheduled on separate days, with one context tested per day (Figure 1A). On each day, participants first completed a value learning task followed by a WM delay-estimation task under the same context. The order of the contexts was counterbalanced across subjects (n=14, power analysis see A.1.2). In the baseline context, all orientations were associated with the same reward; in the reward context, reward increased with the diagonality of the orientations (Figure 1B). We designed the reward context such that the reward structure goes against the environmental distribution of the orientations (Figure 1C), where cardinal orientations are more prevalent [Girshick et al., 2011]. This allowed us to disentangle the impact of the environmental prior and learned reward.

In the value-learning task (Figure 1D), each trial started with four gratings with orientations sampled from the environmental distribution presented in different quadrants on the screen. Participants used mouse clicks to choose one grating and received feedback of the rewards (indicated as points). They were instructed to maximize the reward while learning the reward-orientation association. Each participant completed 5 blocks of 20 trials. At the end of the training, participants learned to choose the most diagonal grating in the reward context, while they choose the most diagonal one at a chance level in the baseline context (Figure 1D, details of learning trajectory in A.1.1).

After the value-learning task, participants performed the WM delay-estimation task (Figure 1E). Participants viewed a centrally presented grating (the target, 0.5s duration), and reproduced its orientation after a 1s or 5s delay. Orientations of the targets in each run were pseudo-randomly drawn from a uniform distribution. The rewards (points) they earned on each trial decreased linearly with the absolute memory error (the reported orientation minus the true orientation) in both contexts. The orientation of the target determined the maximum reward (when memory error = 0°) that can be earned in each trial. Following the same policy they just learned (Figure 1B), in the reward context, more diagonal targets were associated with higher reward, while the rewards were uniform across

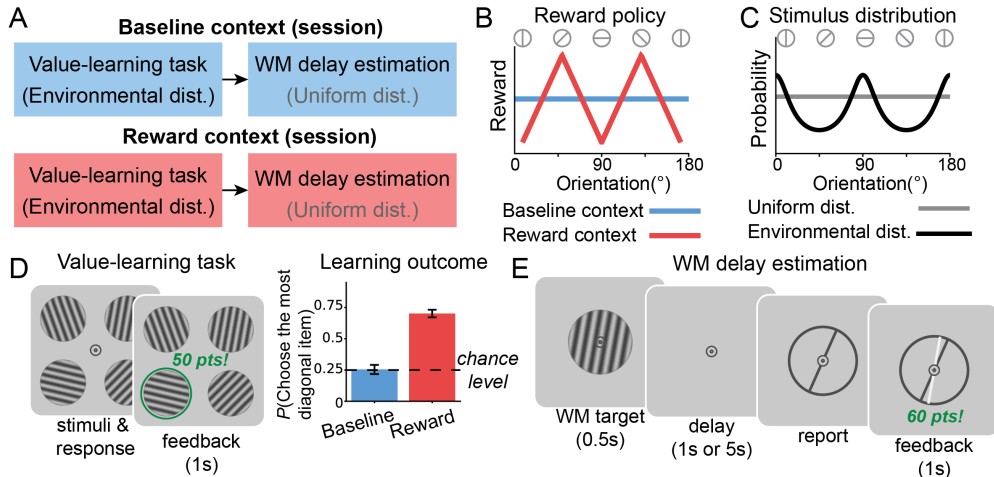

Figure 1: Behavioral experiments and procedures. (A) The experiments included 2 days, each day assigned to a distinct reward policy. (B) Reward policy for the baseline and the reward context. (C) Uniform distribution and environmental distribution of orientations adopted from Girshick et al. [2011]. (D) Left: The value-learning task: participants learn the reward-orientation association in a 4AFC decision task. Right: The learning outcome of the value-learning task. Participants learned to choose the most diagonal orientation in the reward context, while the choice is random in the baseline context. (E) The WM delay-estimation task, where the memory of a single orientation is tested after a 1s or 5s delay.

orientations in the baseline context. By the end of each session (day), the total rewards (points) earned across the two tasks were converted into cash bonus earned by the participants.

## 2.2 Behavioral results

### Reward stabilizes WM representations

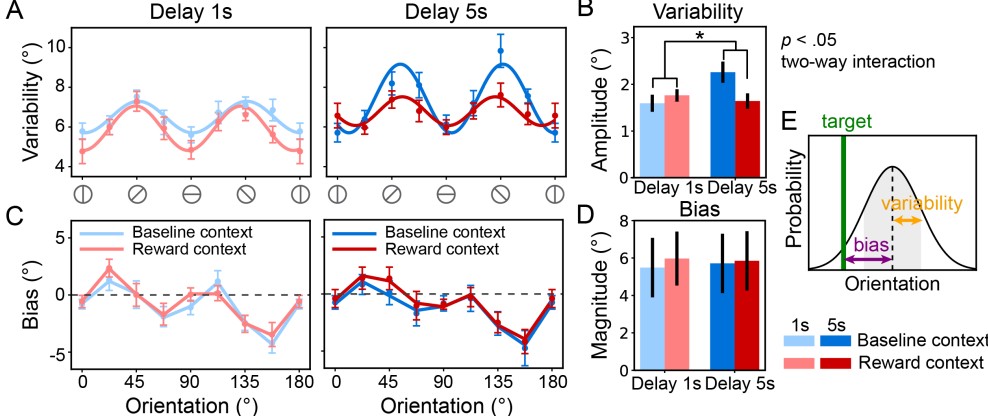

Figure 2: Behavioral results. (A) WM variability (bias removed) in each context x delay condition. Data points represent mean ± 1 s.e.m. Lines are fitted variability (sinusoidal) functions. (B) Variability amplitude fitted from sinusoidal functions. Higher amplitude indicates a stronger oblique effect—cardinals having a lower variability compared to the oblique orientations. Error bars represent ± 1 s.e.m. (C) WM bias in each context x delay condition. Data points represent mean ± 1 s.e.m. Lines are fitted polynomial functions. (D) Bias magnitude estimated from polynomial function, where higher value denotes a higher repulsion away from the cardinals. Error bars represent ± 1 s.e.m. (E) Illustrations of how bias and variability are decomposed from memory responses.

It is well-established that orientation estimation exhibits distinct patterns of bias and variability [Taylor and Bays, 2018, Wei and Stocker, 2017, Girshick et al., 2011]. We decomposed the memory error into these two components (Figure 2E). For each context x delay condition, we fitted polynomial regression to the (signed) memory error to predict the bias. After subtracting the predicted bias from the error, we used the standard deviation of the residuals as the variability (details in A.1.5). For visualization, the results were binned based on the orientations of the WM target in Figure 2.

For the variability, we found a significant 3-way interaction indicated by a linear mixed-effects model (context x delay x orientation, $p < .05$; Figure 2A). In the short delay, there was a well-known "oblique effect" [Appelle, 1972, Girshick et al., 2011, De Gardelle et al., 2010]: the variability was lower for cardinal compared to the diagonal orientations in both contexts (no interaction of context and orientation, $p = .16$). In the long delay (5s), there was a significant interaction of context and orientation ($p < .05$), where the variability of the diagonal orientations was lower in the reward context compared to the baseline. That is, after value learning, high-reward (diagonal) orientations were maintained better, reflecting the effect of learned stimulus-reward association on stabilizing WM. Interestingly, When the reward is uniform, the oblique effect became more pronounced in the long delay, while the cardinal orientations were more stable (also see Figure 3A, where the variability is grouped by contexts). Therefore, more prevalent stimuli in the environment not only were encoded more precisely, but maintained better in WM. In contrast, in the reward context, the oblique effect was attenuated over time, due to the reward structure that opposed the environmental prior. Overall, our results reveal a joint effect of reward and natural stimulus prior on the variability of WM.

We further confirmed the above effects by fitting the variability pattern by sinusoidal functions over orientations (details in A.1.6). The parameter "oblique amplitude" captured the relative advantage in variability at cardinal compared to the oblique orientations (Figure 2B). As a result, we found a significant 2-way interaction of context and delay ($p < .05$), and no main effect of context ($p = .19$) nor delay ($p = .20$).

For the bias, we observed a repulsion bias, with the reported orientations shifted away from the cardinal orientations [Taylor and Bays, 2018, Wei and Stocker, 2017, Girshick et al., 2011]. A linear mixed-effect model showed no significant 3-way interaction between context, delay, and orientation ($p = .09$, Figure 2C), and the bias pattern was not affected by either the delay or the reward context. We further quantified the bias magnitude (Figure 2D, details in A.1.4), and found no interaction between context and delay ($p = .78$), nor any main effect of context ($p = .75$) or delay ($p = .95$). Given that reward and delay predominantly affected variability, below we focused on modeling their effects on memory variability.

**WM resource allocation shaped by prior and reward**

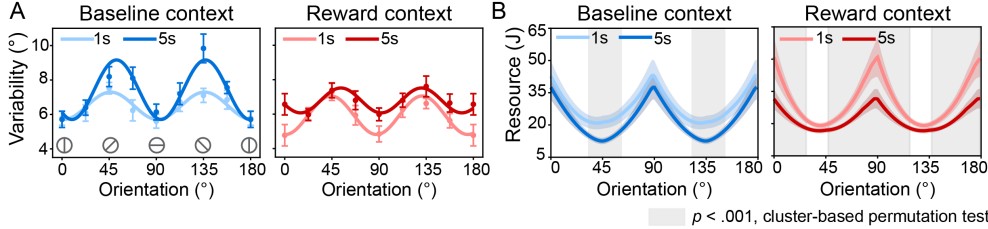

Figure 3: The distribution of behavioral variability and estimated resource $J$. (A) WM variability grouped by the context. (B) The resource distribution estimated by a variable precision model. The shaded areas represent the clusters showing significant differences in resource between the two delay durations (permutation test, $p < .001$). Color shades and error bars represent ± 1 s.e.m.

We view WM resource as quantities that can be distributed to improve encoding and maintenance quality. From behavioral data, we estimated the theoretical resource quantity using Fisher information $J$ by the Variable Precision (VP) model [Van den Berg et al., 2012]. In this model, the magnitude of memory error is associated with WM resource $J$. We extended the VP model by allowing the resource $J$ to vary as a function of orientation, and quantified how the resulting resource profiles changed systematically across contexts and delays (model details and parameter tests in A.1.7).

We fitted the VP model to individual data in different contexts and delays, and then we treated the best-fit resource function $\bar{J}(\theta)$ as the estimated distribution of WM resource (Figure 3B). The estimated resource roughly followed the environmental distribution of orientations, peaking at cardinal orientations. Critically, rewards affected how WM resource evolved across delay. When reward was uniform (baseline context), resource declined over time for diagonal orientations but were maintained well for cardinal ones, reflecting preferential maintenance for more frequent stimuli. In contrast, in the reward context, the cardinal orientations (now associated with lower reward) have their resource decreased under longer delay whereas the resource for the diagonal orientations (associated with higher reward) was stable. These results confirm a joint effect of stimulus prior and rewards on WM resource allocation.

## 3 Efficient coding for utility maximization in working memory

### 3.1 Previous studies on efficient coding of perception

The theory of efficient coding posits that perceptual system should allocate more encoding resource with respect to the statistical structure of the environment by allocating more resource to more common stimuli [Barlow et al., 1961]. Recent studies on efficient coding have derived optimal allocation of neural resource quantified as Fisher information [Wei and Stocker, 2015, Hahn and Wei, 2024, Morais and Pillow, 2018]. Under this framework, resource allocation can be formalized as the objective to minimize overall expected loss defined as:

$$L = \int p(\theta) \, \mathbb{E}\left[ |\hat{\theta}(m) - \theta|^p \, \Big| \, \theta \right] d\theta \tag{1}$$

Here $m$ is the sensory measurement of the stimuli, and $p$ is an exponent term that specifies the loss function. Extending the theoretical framework developed in Wei and Stocker [2015], Morais and Pillow [2018] linked the estimation error with neural resource (Fisher information $J$), and identified the lower bound of the expected loss in Eq.1 above as $L = \int p(\theta) \, J(\theta)^{-\frac{p}{2}} \, d\theta$. Under the constraint of total resource $\int J(\theta)^\beta \, d\theta < C$, where $C$ represents total resource and $\beta$ specifies the non-linearity in the constraint, it was shown that the optimal resource allocation is proportional to the stimulus prior distribution raised to a power $q$, a power-law efficient code [Morais and Pillow, 2018]:

$$J_{\text{opt}} \propto p(\theta)^q$$

### 3.2 Extension of power-law efficient code to utility maximization

Here, we extended this framework to tasks in which stimuli $\theta$ carry context-dependent rewards. Similar ideas have been proposed for perceptual encoding [Schaffner et al., 2023]. We shift the objective from minimizing estimation errors to maximizing expected utility. In our tasks, reward decreases monotonically with estimation error. Therefore, the expected reward can be written as:

$$R = \int p(\theta) \, v(\theta) \, \mathbb{E}[d(\hat{\theta}, \theta)] \, d\theta$$

where $v(\theta)$ is the value function determining the orientation-reward association, $d(\hat{\theta}, \theta)$ is a reward discounting function—a decreasing function of error. A simple choice is $d(\hat{\theta}, \theta) = \beta - c|\hat{\theta} - \theta|$, where $\beta > 0$ can be considered as a baseline payoff (e.g., the subject payment), and $c > 0$ dictates how rapidly rewards declines with estimation error. Given that $\beta$ and $c$ are constants, maximizing reward is equivalent to minimizing the loss:

$$L = \int p(\theta) \, v(\theta) \, \mathbb{E}|\hat{\theta} - \theta| \, d\theta$$

This objective has a form similar to Eq.1. Thus, the power-law efficient coding can be extended for utility maximization as:

$$J_{\text{opt}} \propto [p(\theta) \, v(\theta)]^q \tag{2}$$

That is, the optimal resource is proportional to the expected value of the stimulus raised to a power. In the reward context of our behavioral experiments, reward increased linearly with the diagonality of

orientations, opposite to the trend in natural stimulus statistics. Consequently, the optimal resource allocation, and thus the variability of perceptual or memory reports, would be flattened relative to the baseline condition where reward is uniform (Figure 4A). These predictions are in line with our behavioral results (Figure 4B).

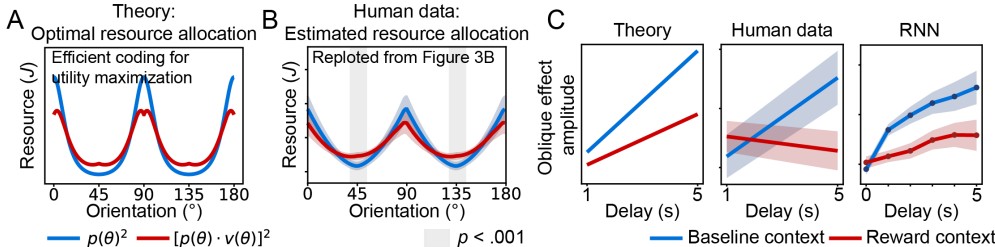

Figure 4: Utility maximization extension of efficient coding theory. (A) Optimal resource $J$ for baseline context ($J_{\text{opt}} \propto p(\theta)^q$), and reward context ($J_{\text{opt}} \propto [p(\theta)\,v(\theta)]^q$). Here set $q = 2$ for illustration. (B) Resource allocation estimated from the behavioral data (5-sec delay) was consistent with the theoretical prediction (A). The shaded gray areas represent the clusters showing significant differences in resource between the two contexts (permutation test, $p < .001$). (C) Left: The optimal resource allocation predicts a stronger oblique effect in the baseline than in the reward context. In addition, the increment of the oblique effect over time is more rapid in the baseline context. These predictions are consistent with the behavioral data (middle) and the RNNs (right). Color shade error bar as the standard error.

## 3.3    Considerations on the time domain

One finding in the behavioral data was that, in the baseline context, the oblique effect became stronger with a longer delay. This indicates that less frequent stimuli are not only encoded with lower precision, but are also harder to maintain over time. Here, we extend the power-law efficient code to incorporate the time domain, allowing resource to vary with delay. The objective is to optimize $J(\theta, t)$, the resource distribution over orientations and time by minimizing the loss function:

$$\int_0^T \int p(\theta)\, J(\theta, t)^{-\frac{p}{2}}\, d\theta\, dt$$

subject to the resource constraints summed over time:

$$\int_0^T \int J(\theta, t)^{\beta}\, d\theta\, dt < C$$

Let $\lambda$ be the Lagrange multiplier. The full Lagrangian is:

$$\mathcal{L} = \int_0^T \int p(\theta)\, J(\theta, t)^{-\frac{p}{2}}\, d\theta\, dt + \lambda \left( C - \int_0^T \int J(\theta, t)^{\beta}\, d\theta\, dt \right)$$

Solving for $J(\theta, t)$ yields:

$$J_{\text{opt}}(\theta, t) \propto p(\theta)^q$$

Importantly, even when incorporating the time domain, the optimal distribution of resource $J_{\text{opt}}$ depends only on the stimulus distribution $p(\theta)$ regardless of time, implying a fixed strategy throughout the memory delay. As a consequence, the *difference* in resource allocated between frequent (cardinal orientations) versus infrequent (diagonal orientations) stimuli will accumulate over time, producing a stronger oblique effect with longer delays, consistent with our empirical results (Figure 4C in baseline context). This stable strategy can also be applied to utility maximization in which the optimal solution shared the same form as Eq. 2.

## 4    Recurrent neural network

The neural dynamics of WM has been extensively studied and modeled by recurrent neural networks (RNNs) with attractor dynamics that hold memory content in the presence of neural noise [Amit and

Brunel, 1997, Bouchacourt and Buschman, 2019, Compte et al., 2000, Wang, 1999, 2021, Wimmer et al., 2014, Esnaola-Acebes et al., 2022]. Here, we investigated whether and how biologically plausible circuit architecture RNNs, when combined with task-oriented and simple learning rules, can implement efficient and adaptive allocation of WM resource to maximize expected reward, and how their behaviors compared to human performance.[1].

## 4.1 RNN setup

We trained RNNs to perform the same WM delayed-estimation task as humans (Figure 1E). To systematically investigate the effect of prior and reward, we trained the RNNs in a 2x2 design (Figure 5A). For the prior, input orientations were sampled either from a uniform distribution or a natural (environmental) distribution, yielding the Uniform RNN and Environmental RNN, respectively. For the reward, networks were trained to maximize reward under two contexts (implemented by the loss functions): a baseline context with uniform reward across orientations and a reward context where reward increased with orientation diagonality.

### Architecture

The input orientation $\theta \in [0, \pi)$ was encoded by 32 neurons with circular-normal tuning function $u_i(\theta) = \exp\left[\kappa_{\text{in}} \cos\left(2(\theta - \phi_i)\right)\right]$, which were later normalized to $[0, 1]$. The centers of the encoding tuning functions $\phi_i$ evenly tiled the orientation space $[0, \pi)$. On each trial, an orientation stimulus was presented for 0.5s, followed by a 5s memory delay (matched the human WM task) during which neural activity was maintained through recurrent connections among the neurons governed by $W_{rec}$.

The activity of the recurrent layer was updated based on the equation:
$$r(t+1) = (1-\alpha)\, r(t) + \alpha\, \tanh\left(r(t)W_{\text{rec}} + u(\theta) + \sigma\, \xi(t)\right)$$
where $\alpha$ was set as 0.2. The neural noise was added at each time step (per 20 ms) as $\xi(t) \sim \mathcal{N}(0, I)$, with a magnitude $\sigma = 0.5$.

The outputs of the 32 response neurons are linear readout from the recurrent $r(t)$ activity as $\hat{z}(t) = W_{\text{out}}\, r(t)$. Each output unit $\hat{z}_i(t)$ corresponded to an orientation centered at the same preferred angle $\phi_i$ as its encoding counterpart, allowing the estimated orientation $\hat{\theta}$ to be decoded via population vector averaging.

### Loss functions and training

The RNNs were trained to minimize the loss function
$$\mathcal{L} = \frac{1}{T}\sum_{t=1}^{T} v(\theta)\left\|\Delta\theta(t)\right\|^2 \; + \; \lambda_{\text{act}}\, \frac{1}{T}\sum_{t=1}^{T}\|r(t)\|^2 \; + \; \lambda_w\, \left\|W_{\text{rec}}\right\|^2$$

where $\left\|\Delta\theta(t)\right\|^2$ denotes the squared estimation error between the stimulus orientation $\theta$ and estimated orientation $\hat{\theta}$ at time step $t$. The total time steps per trial is $T$. The function $v(\theta)$ specifies the stimulus-reward association. Under the baseline context, reward was uniform across orientations so this term was negligible, and was set as $v(\theta) = 1$. Under the reward context, reward increased with orientation diagonality (Figure 5A), computed as:
$$m(\theta) \;=\; \theta \bmod \frac{\pi}{2}, \quad d(\theta) \;=\; \frac{\pi}{4} \;-\; \left|m(\theta) - \tfrac{\pi}{4}\right|, \quad v(\theta) \;=\; 0.5 \;+\; \frac{d(\theta)}{\pi/4}$$

This weighting makes errors on more rewarded orientations contribute more strongly to the loss. In addition to the estimation error and reward, we imposed $L2$ regularization on recurrent activities ($\lambda_{\text{act}} = 10^{-4}$), and weights ($\lambda_w = 10^{-4}$) to mimic biological resource constraints. In addition to different reward contexts, the networks were also trained under different stimulus priors by sampling the orientation $\theta$ either from a uniform or from the environmental prior during the training (Figure 5A). We trained the networks using the Adams optimizer implemented in PyTorch with a learning rate of $10^{-3}$. Each network reported here was trained with 15 random initializations through 2000 epochs with batch size as 256. After training, each network was tested with orientations uniformly sampled from 0° to 180° in steps of 2°, with 300 repetitions per orientation, and we reported the results averaged over all the initializations.

---

[1]Scripts are available at `https://github.com/Qingqing-Yang-177/wm_UtilityMax.git`

## 4.2 RNNs results

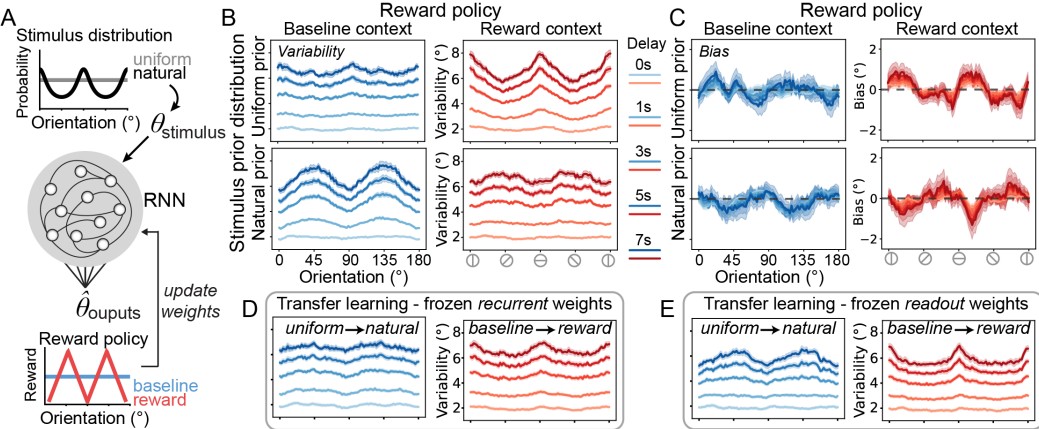

Figure 5: RNNs architecture, and their variability and bias patterns. (A) RNN architecture, training stimulus distribution (uniform and natural distribution) and loss functions (reward and baseline context). (B) WM variability from RNNs trained with each stimulus distribution x context condition. (C) Bias of RNNs trained with each stimulus distribution x context condition. (D) Transfer learning results when we froze recurrent weights trained in uniform distribution + baseline context, and retrained the readout layer in the new conditions. (E) Transfer learning results when we froze readout layer trained in uniform distribution + baseline context, and retrained the recurrent weights in the new conditions. Only the results of variability are shown here. Each line denotes a delay length after the stimulus presentation duration. Color shades represent the standard error.

We trained the RNNs under different stimulus priors and reward contexts. Bias and variability were quantified using the same procedure as for human data. RNNs trained with the natural stimulus distribution and uniform reward (baseline context) showed the oblique effect: memory variability was lower for cardinal than diagonal orientations (Figure 5B natural distribution + baseline context), consistent with prior work using biased training stimuli in RNNs [Xiong and Wei, 2022, Ye et al., 2025, Eissa and Kilpatrick, 2023]. We found that with longer delays, variability increased more for infrequent (diagonal) orientations, amplifying the oblique effect, mirroring the behavioral data in the baseline context (Figure 3A).

When RNNs were trained with natural prior but with a reward policy favoring diagonal orientations, higher rewards stabilized these orientations by reducing their variability (Figure 5B natural distribution + reward context). Because the reward policy opposed the natural stimulus prior, their joint influences led to a weakened oblique effect. These RNN patterns are in line with efficient, utility-maximizing allocation of WM resource (Figure 4C), and resemble the behavioral results especially in the longer (5s) delay, where the oblique effect was significantly attenuated in the reward context relative to the baseline context (Figure2A).

We also trained RNNs with uniform stimulus distribution under both contexts. As expected, when both prior and reward were uniform (Figure 5B uniform distribution + baseline context), variability was flat across orientations and simply increased over time due to the accumulated neural noise. Although human data did not show as strong a reward effect, the results from uniform prior and reward context isolated the influence of reward on memory variability (Figure 5B uniform distribution + reward context). In this case, memory variability was lower for higher-reward orientations (diagonal) than lower-reward orientations (cardinal), and this pattern grew with delay.

While the RNNs reproduced clear effects of stimulus prior and reward on WM variability, and consistent with the human data, the bias patterns were less systematic (Figure 5C). A slight repulsion appeared in the RNNs trained with a uniform prior under the reward context; however, this pattern likely reflects an attraction toward the high-reward orientations rather than the repulsion as in the previous studies on orientation Girshick et al. [2011], Wei and Stocker [2017]. For RNNs trained with natural stimulus distribution with baseline context, we did not observe a clear repulsive bias (from cardinal) as seen in humans, and there was a trend of shifting towards more frequent stimuli.

### 4.3 Robustness and generalization

We performed systematic testing of the RNNs by varying network size (32 vs. 128 units), the internal recurrent noise level, the tuning centers of the encoding neurons, and regularization terms (details in A.2.4). These analyses showed that the effects of reward and prior on working memory variability were robust across architectures and configurations.

To address the concern that RNNs might simply memorize input–output mappings, we trained the RNNs with a delay up to 5 sec, and showed that they generalized to longer (untrained) time points (7 sec) while maintaining the robust prior/reward effects (see Delay 7s in Figure 5). These results confirmed that the models relied on dynamic memory representations, not static pattern matching.

### 4.4 Dissecting the contributions of recurrent dynamics

To investigate the unique contributions of the recurrent layer and readout layer, we performed two sets of transfer learning experiments to transfer across (i) prior stimulus distribution and (ii) the contexts, while freezing either the recurrent or the readout layer (details in A.2.2).

We first trained the RNNs under the uniform + baseline condition. In the first experiment, we then froze the recurrent weights, and retrained the readout layer in the (i) natural + baseline or (ii) uniform + reward condition. We observed a weaker effect from both prior and reward (Figure 5D): The variability patterns remained relatively flat across orientations compared to the RNNs fully trained in the new conditions (Figure 5B). In the second experiment, we instead froze the readout layer, and retrained the recurrent layer. We found that the variability patterns (Figure 5E) were more similar to the RNNs fully trained in the new conditions. Therefore, the recurrent dynamics plays a more important role in allowing the RNNs to adapt to different stimulus distributions or reward policies.

### 4.5 Input noise and bias

While the bias was less systematic in our RNNs, there was weak but noticeable attraction bias toward the high prior or high reward stimuli (Figure 5C). The "*attraction to prior*" effect has been found in some WM studies that manipulated stimulus prior distributions [Panichello et al., 2019, Honig et al., 2020], and is generally in line with the Bayesian theory that an ideal observer would exhibit attraction to the prior [Knill and Pouget, 2004, Ma, 2019]. The repulsive (from cardinal) bias often observed in orientation estimation (and other domains) is counterintuitive and of interest in recent development on efficient coding [Wei and Stocker, 2015, 2017, Hahn and Wei, 2024]. One possible account is that repulsion arises in an early encoding stage, where less frequent stimuli are represented with greater noise, preceding the recurrent network we modeled here. Thus, we trained a variant of RNNs where we injected input noise to the stimuli, with magnitude varying inversely with the environmental prior (stronger noise for more diagonal orientations) during training [Gu et al., 2025]. The model exhibited a clear repulsive bias away from the cardinals (Figure 6, and details in A.2.3).

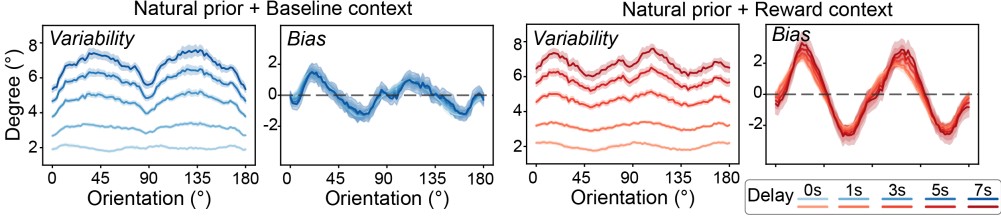

Figure 6: RNNs variability and bias when trained with non-homogeneous input noise. Larger input noise near diagonal orientations produced a repulsive bias.

## 5 Limitations

While a repulsive bias could emerge when inhomogeneous input noise was introduced, such implementation does not explain how such encoding could be learned by neural networks. A more complete account may require training a hierarchical model, or multi-layer RNNs [Yang et al., 2024],

with a dedicated trainable encoding and maintenance stage to capture how the repulsive bias emerge through learning.

In this study, we used gradient-descent and backpropagation through time as the learning algorithms and showed that RNNs can learn to efficiently allocate resource for utility. Future studies can explore whether the same model behaviors emerge under more biologically plausible learning rules [Miconi, 2017, Bellec et al., 2020].

## 6 Discussions

Using a novel behavioral paradigm, we demonstrated that reward influences the precision and temporal dynamics of visual WM. Once stimulus-reward associations were learned, memory representations for high-reward stimuli were maintained with greater stability over time, reflected in the lower memory variability. These findings contrast with previous studies reporting null or mixed effects of monetary reward on WM [Van den Berg et al., 2023, Brissenden et al., 2023, Manga et al., 2020, Weiss et al., 2025]. The critical difference may lie in how the reward was manipulated: previous studies varied reward cues randomly across trials, whereas our design established consistent stimulus–reward associations that had to be learned. Treating reward as a stable feature of the environment may elicit a more robust value-driven influence on WM, as in natural environments where reward contingencies are learned through experience.

By treating stimulus–reward associations as an integral part of the environment, we extended efficient coding theory [Wei and Stocker, 2015, Morais and Pillow, 2018] from minimizing estimation errors to maximizing expected utility, and further incorporated the temporal dimension for WM. The optimal resource allocation respects stimulus probability and associated reward jointly, and this optimal strategy remains stable over time. Importantly, both human behaviors and RNNs aligned with these theoretical predictions: (1) showing reduced oblique effect when diagonal orientations carried higher rewards, and (2) the effect of natural stimulus prior and the reward both accumulated over time. These converging results suggest that optimizing resource allocation with respect to utility, beyond merely minimizing estimation error, is a fundamental component of adaptive behavior. To do so, organisms efficiently allocate WM resource to stabilize highly rewarding WM representations over time.

Previous studies on efficient coding jointly explained perceptual bias and discrimination threshold [Wei and Stocker, 2015, 2017], and suggested a coupling between the two. Our normative model and behavioral data diverged from this view: In extending efficient coding theory to the temporal domain and reward maximization, we focused on linking neural resource $J$ with variability, without specifying the direction of bias. In terms of behavioral results, while variability increased over time, bias remained relatively stable, aligning with previous WM studies on orientations [Tomić et al., 2025, Shin et al., 2017]. The temporal dissociation between them may indicate a distinction between perceptual encoding and memory maintenance. Two factors may contribute this dissociation: First, this interpretation is consistent with findings that in human brains, WM engages neural populations that go beyond initial sensory encoding [Christophel et al., 2017, Li et al., 2021, Li and Curtis, 2023, Li et al., 2025]. While encoding depends on sensory areas with fixed tuning curves, WM further involves higher-level regions including parietal and prefrontal cortices, where neurons exhibit mixed and flexible selectivity [Rigotti et al., 2013, Fusi et al., 2016]. The dynamics of the higher-level regions may differ from the original encoding structure. Second, recent theoretical work [Hahn and Wei, 2024] suggests that bias arises from two opposing forces: a repulsive component during encoding and an attractive component during Bayesian decoding. These components can coexist, and if both strengthen over the delay, their effects may offset each other, resulting in a stable overall bias.

To conclude, the present study provides empirical evidence that humans allocate WM resource according to the expected utility from the learned stimulus-reward associations. These behaviors are consistent with an efficient coding framework in which the objective is to maximize individual utility. Notably, similar patterns emerge in biologically-plausible RNN trained with a simple learning rule to maximize expected reward. Together, our behavioral results, neural network models, and theoretical framework revealed that modulations in memory variability serve as the primary mechanism through which the WM system allocates resource based on stimulus statistics and reward. Beyond their relevance to biological systems, our findings indicate that designing artificial agents aligned with human adaptive behavior requires integrating motivational and environmental factors that support utility maximization, rather than focusing solely on minimizing errors.

## Acknowledgments and Disclosure of Funding

The authors thank the anonymous reviewers for their constructive feedback during the review process.

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

# A  Technical Appendices and Supplementary Material

## A.1  Behavioral methods and results

### A.1.1  Value learning trajectory

The value learning task followed the reward-orientation association that, in the reward context, reward increased with the diagonality of the orientations; in the baseline context, all the orientations were associated with the same reward (Figure 1B). To visualize the learning trajectory, trials were binned into 10 trials per bin based on their order (Figure 7). The percentage of subjects choosing the most diagonal stimuli in the reward context was consistently higher than in the baseline context after 40 trials of learning (FDR corrected paired t-test, $p < .01$). Learning in the reward context reached an asymptote (about 70% choosing the highest-reward stimuli) at around 60 trials. In the baseline context, the probability of choosing the most diagonal stimuli (now not high-reward) of all participants reached the chance level 25% after 70 trials of learning (FDR corrected one-sample t-tests, $p > .05$).

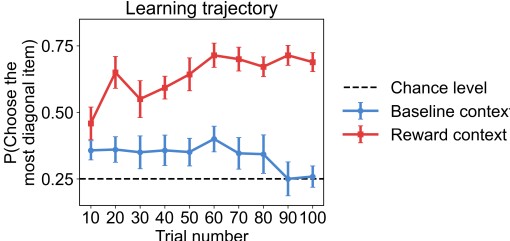

Figure 7: Learning trajectory in the value learning task.

### A.1.2  Power analysis

We conducted a power analysis based on the effect size reported in the Exp.1 in Schaffner et al. [2023], which investigated the effect of reward on perceptual encoding. Specifically, we used the reported coefficient (Estimate $= -0.16$, $SE = 0.09$) from the Bayesian hierarchical linear regression model to approximate a standardized effect size (Cohen's $f \approx 2.0$). This effect size represents the magnitude of the reward-related difference between the reward and baseline contexts for oblique orientations. Based on the orientation bins, we applied a Bonferroni correction for multiple comparisons, resulting in a corrected significance threshold of $\alpha = 0.01$. Power analysis revealed that 14 participants provided above 90% power to detect an effect of reward in the oblique orientation.

### A.1.3  Compensation information

Each subject was compensated at the rate of 12 dollars per hour for each session, and there was up to 10-dollar bonus based on the points collected in each day (context).

### A.1.4  WM bias magnitude

We modeled the signed response $error$ as a periodic function of the stimulus orientation $\theta \in [0°, 180°]$ using a third-order Fourier series regression. For each context $\times$ delay condition, the model was defined as:

$$\hat{error}(\theta) = \beta_0 + \sum_{j=1}^{3} [\beta_{c,j} \cos(j\theta) + \beta_{s,j} \sin(j\theta)] + \varepsilon,$$

which approximates the response error as a weighted sum of sinusoidal functions. Here the free parameters included $\beta_0$ as the intercept, and $\beta_{c,j}$ and $\beta_{s,j}$ as the coefficients (weights). All parameters were estimated via ordinary least squares by minimizing the sum of squared difference between predicted $\hat{error}(\theta)$ (the bias) and observed $error(\theta)$.

To quantify the overall magnitude of the bias, we compared the signed peaks within 0–45° ($\hat{error}_1$) and within 135–180° ($\hat{error}_2$), each determined by the maximum absolute value of the predicted bias in that range. The resulting bias magnitude $M$ was computed as

$$M = \hat{error}_1 - \hat{error}_2.$$

A positive bias magnitude represents a repulsive bias from the cardinal orientation, and a negative value denotes an attraction towards cardinals. If $\hat{error}_1$ and $\hat{error}_2$ share the same sign, the corresponding value was excluded from group analyses.

### A.1.5 WM variability

To isolates trial-by-trial variability independent of systematic, orientation-dependent bias, the predicted bias $err\hat{o}r(\theta)$ from Fourier series regression was subtracted from the response error to yield a bias-removed residual:

$$error(\theta)' = error(\theta) - err\hat{o}r(\theta),$$

We quantified the WM variability $Var(\theta)$ within each condition (and orientation bin) as the standard deviation of the residual $error(\theta)'$, providing a bias-removed response precision.

### A.1.6 Quantify effects in WM variability

After obtaining bias-removed variability $Var(\theta)$, we quantified the effect of context $\times$ delay in variability by fitting a sinusoidal function of the form:

$$\hat{Var}(\theta) = A_{var} \sin\left(\frac{\omega\theta\pi}{180} + \phi\right) + Base_{var},$$

where $\theta$ denotes the orientation in degrees, $\omega$ controls the frequency, $\phi$ the phase offset, $A_{var}$ the amplitude, and $Base_{var}$ the baseline level of variability which is constant across orientations. This formulation captures systematic modulations in response variability across orientations (e.g., oblique effects).

Model parameters $(\omega, \phi, A_{var}, Base_{var})$ were estimated separately for each condition by minimizing the squared error between the observed and predicted variability:

$$\min_{(\omega,\phi,A,B)} \sum_{\theta} \left[Var(\theta) - \hat{Var}(\theta)\right]^2,$$

using a global optimization approach with randomized initialization and bounded parameter ranges to ensure robust convergence.

Among the parameters, the amplitude $A_{var}$ reflects the strength of modulation in variability from orientation. Therefore, we use $A_{var}$ to capture the relative advantage in error variability at cardinal orientations, where a larger $A_{var}$ indicate a higher difference in variability for cardinal relative to oblique orientations, such as a stronger oblique effect.

We fitted each participant's variability results in each context x delay condition with the sinusoidal function above, and applied repeated-measured ANOVA to investigate the effect of reward and context:

**For $A_{var}$**, the amplitude parameter (Figure 2B), there was a significant interaction of context and delay ($p < .05$), and no main effect of context ($p = .19$) nor delay ($p = .20$). The strength of the oblique effect - how much the cardinal orientations has lower variability compared to the oblique - was affected by the reward-orientation association in the WM longer delay.

**For $Base_{var}$**, the base level of the variability across orientations, there was no significant interaction of context and delay ($p = .76$), but a significant main effect of context ($p = .04$) and delay ($p < .001$).

**For $\phi$**, the peak location, there was no significant interaction of context and delay ($p = .39$), and no main effect of context ($p = .25$) and delay ($p = .41$). The peak was stable around the oblique orientations.

**For $\omega$**, the frequency, there was no significant interaction of context and delay ($p = .57$), and no main effect of context ($p = .13$) and delay ($p = .22$). The frequency was stable to provide a $\sim 90°$ periodicity.

### A.1.7 WM resource allocation distribution

From the behavioral data, we estimated the theoretical resource quantity as Fisher information $J$. We adapted the Variable Precision (VP) model to fit memory error distributions under different contexts and delays [Van den Berg et al., 2012]. Memory errors (bias-removed) were modeled as samples from von Mises distributions $p(\epsilon) = \texttt{von\_Mises}(0, \kappa)$, where $\epsilon$ denotes the angular memory error and $\kappa$ is the concentration parameter. In the model, $\kappa$ was linked with the fisher information — WM resource parameter $J$ by $J = \kappa \cdot \frac{I_1(\kappa)}{I_0(\kappa)}$, where $I_0(\kappa)$ and $I_1(\kappa)$ are the modified Bessel functions of the first kind (of order 0 and 1). Higher J values produced narrower distributions. Although not our main focus, the VP model assumed that J itself is a random variable following a gamma distribution $J = \texttt{gamma}\left(\frac{\bar{J}}{\tau}, \tau\right)$, where $\bar{J}$ is the mean of the resource parameter $J$, and $\tau$ is the scale parameter of the Gamma distribution, controlling the variability in $J$ across trials [Van den Berg et al., 2012]. To capture orientation-dependent resource allocation, we modeled the mean resource $\bar{J}$ as a function of orientations:

$$\bar{J}(\theta) = A \cdot (1 - B \cdot |\sin(2\theta + \phi)|)^n$$

, where $A$, $B$, $\phi$ and $n$ were fitted as free parameters, allowing $\bar{J}$ to vary across orientations.

For the resource distribution represented by the best-fitted resource function $\bar{J}(\theta) = A \cdot (1 - B \cdot |\sin(2\theta + \phi)|)^n$, Figure 3B showed that resource distributions exhibited a stronger oblique effect (the $W$ pattern) in the long delay condition in the baseline context, but a weaker oblique effect in the long delay condition in the reward context (a shallower $W$). These effects are reflected in the parameter $B$. Indeed, we found a significant interaction effect (context $\times$ delay, $p < .05$) when testing the best-fitted $B$ values. Here we describe the statistical results on all the fitted parameters:

**For $A$:** The linear mixed-effects model showed no significant interaction effect (context $\times$ delay, $p = .27$), and no significant main effect of context ($p = .27$) or delay ($p = .28$). The Bonferroni-corrected one-sample $t$-test showed that $A$ is significantly larger than 0 ($p < .001$) in all contexts and delays.

**For $B$:** The linear mixed-effects model showed a significant interaction effect (context $\times$ delay, $p < .05$), and a marginally significant effect for delay ($p = .06$) and no effect for context ($p = .5$). The Bonferroni-corrected one-sample $t$-test showed that $B$ is significantly larger than 0 ($p < .001$) in all contexts and delays.

**For $\phi$:** The linear mixed-effects model showed a marginal interaction effect (context $\times$ delay, $p = .06$), and no significant main effect for delay ($p = .22$) or context ($p = .59$). The Bonferroni-corrected one-sample $t$-test showed that $\phi$ is only significantly larger than 0 in the delay $= 5$ s condition in both contexts, while not significantly different from 0 in the delay $= 1$ s condition in either context.

**For $n$:** The linear mixed-effects model showed no significant interaction effect (context $\times$ delay, $p = .25$), but a significant main effect of delay ($p < .05$) and no effect of context ($p = .37$). The Bonferroni-corrected one-sample $t$-test showed that $n$ is significantly larger than 0 ($p < .001$) in all contexts and delays.

## A.2   RNN methods and results

### A.2.1   Computing resource

Modeling training required approximately 10 minutes for each initialization on a single T4 GPU for each setup. Therefore, the results of the main RNN experiment we reported in the paper (Figure 5A) took roughly 10 hours for model training.

### A.2.2   Transfer learning

Additional details of the transfer learning experiments are described here: We conducted 2 transfer learning tasks (across contexts, and priors), and tested the contribution of 2 different layers (recurrent and readout layer) in the RNNs in adapting to stimulus prior distribution and reward policy.

(1) Transfer across contexts (baseline $\rightarrow$ reward). We first trained 15 RNNs in the uniform distribution + baseline context fully with varying initializations, then (i) froze the recurrent weights and retrained the readout in the uniform distribution + reward context; or (ii) froze the readout layer and retrained the recurrent weights in the uniform distribution +reward context. All the other configurations of the RNNs remained the same as those in the main text.

(2) Transfer across prior distributions (uniform $\rightarrow$ natural). We trained 15 RNNs under a uniform distribution + baseline context with varying initializations. After training, we then (i) froze the recurrent weights and retrained only the readout weights in the natural distribution + baseline context; or (ii) froze the readout layer and retrained the recurrent weights in the natural distribution + baseline context. All the other configurations of the RNNs remained the same.

For bias, the transferred RNNs closely matched fully trained RNNs in the new conditions, when we froze and retrain either layer, indicating that both layer explains the effect of prior and reward policy in the bias. The results for the variability are described in the main text (Figure 5D, E ).

### A.2.3   Input noise level

For this RNN variant, we added gaussian noise to the stimulus input during the training while keeping all the other configurations the same. We assumed that the recurrent neurons received a noisy version of the stimulus input. The noisy input $\theta$ was modeled as the true stimulus input $\theta_0$ perturbed by Gaussian noise $\theta \sim \mathcal{N}(\theta_0, \gamma |\sin(2\theta_0)|^2)$, where we set $\gamma$ as 10 [Gu et al., 2025]. This non-homogeneous input noise led to higher input noise for more diagonal orientations. This implementation was based on the assumption that the repulsive bias or oblique effect involves non-homogeneous encoding that occurred before the recurrent neurons. When calculating the estimation error in the loss function, the decoded orientation $\hat{\theta}$ was compared with the true stimulus input $\theta_0$ .

After adding the noise, the stimulus distribution for natural prior condition remained similar to the original stimulus distribution; therefore the RNN training results reflect the effects of both the natural stimulus distribution and input noise (Figure 8). However, the uniform distribution was affected by adding these non-homogeneous

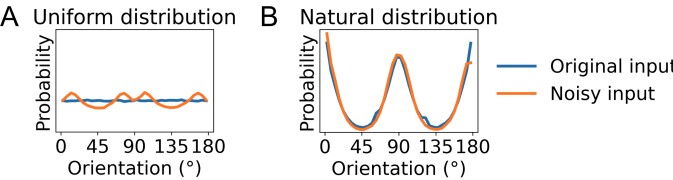

Figure 8: Stimulus distribution before and after adding the stimulus noise.

input noise, and we also observed a decreased variability at oblique orientations compared to the RNNs without input noise.

### A.2.4 Robustness

**Larger network**.

We trained 15 large RNNs with 128 recurrent units (vs. 32 as we reported in the main paper) in each prior x context condition with varying initializations (Figure 9. The variability of the large RNNs under a larger network shared the same pattern with the smaller network, indicating the effect of reward and prior distribution. The average variability was smaller for larger RNNs, suggesting improved precision with larger networks. The magnitude of the bias for larger RNNs was also small.

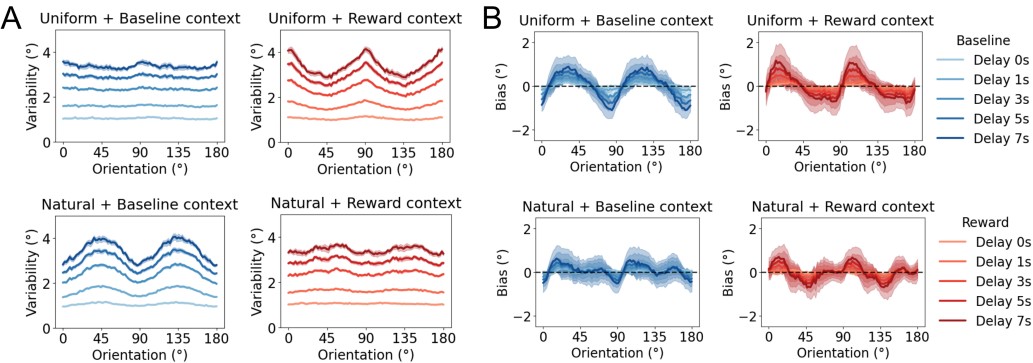

Figure 9: The larger RNN (128 units) WM performance. (A) Variability patterns for larger RNNs reflected the impact of reward and prior stimulus distribution. (B) Bias patterns.

**Alternative loss weightings**.

We trained additional RNNs with larger $L2$ penalty on recurrent weights and activations ($10^{-2}$ vs. $10^{-4}$ as we reported in the main paper), while keeping other configurations the same (Figure 10). We observed that, compared to the weaker regularization, these models with stronger regularization exhibited even stronger modulations by stimulus prior or reward - large differences in variability between low- versus high-probability orientations, or between low- versus high-reward orientations. There was a stronger bias towards the frequent and high-rewarded orientations. Thus, a more constrained working memory system allocates its capacity more selectively across stimuli.

Additionally, we trained RNNs with zero $L2$ penalty on recurrent weights and activations. These models exhibited behavior similar to those trained with small ($10^{-4}$) $L2$ penalties that are affected by both prior and reward conditions. These results suggest that resource-efficient behavior can emerge from task optimization alone, even in the absence of hard-coded limitations. Thus, "limited resource" in working memory may reflect rational strategy to task goals rather than strict architectural bottlenecks.

**Centers of the encoding tuning function**

To investigate whether the results we observed were related to the location of the orientation tuning function of the RNN neurons during stimulus encoding, we trained a collection of RNNs with slightly shifted tuning functions. For each of the 15 RNNs with varying initializations, the unit centers were initialized as

$$centers_i = (\tilde{centers}_i + \phi_{center} + \epsilon_i) \bmod \pi,$$

where $\tilde{centers}_i$ are fixed center values that evenly spaced the orientation space, $\phi_{center}$ is a global phase shift uniformly sampled from $(-5°, 5°)$. The $\pm5°$ shift range corresponds approximately to one spacing step

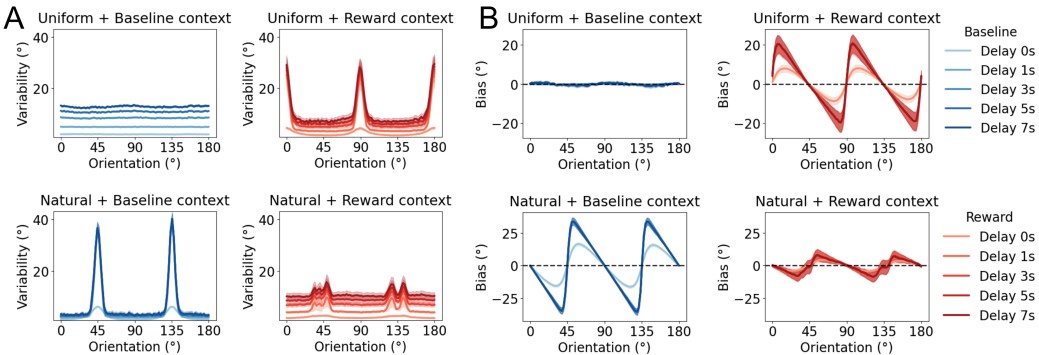

Figure 10: WM variability and bias for RNNs with larger $L2$ regularization on recurrent weights and activations.

$(180°/32)$ between adjacent orientation-tuned units. The $\epsilon_i \sim \mathcal{N}(0, 0.5°)$ is a small per-unit jitter. All other aspects were identical to the original models. The resulting RNNs behaviors were the same as what we reported in the main paper (Figure 11).

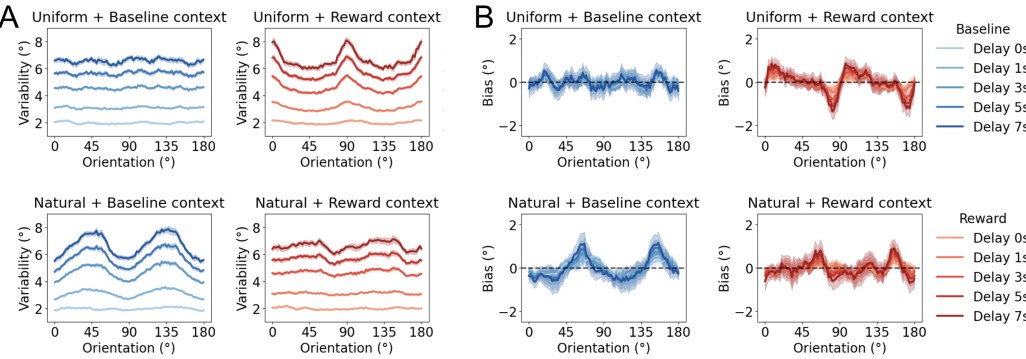

Figure 11: WM performance for RNNs with the shifted centers of the tuning functions during perceptual encoding.

**Internal noise level**

To investigate the effect of internal noise, we trained and tested RNN under different levels of internal sensory noise levels ($\sigma = 0.1 - 0.9$, the neural noise added to the recurrent activation per time step), as shown in Figure 12A. (I) When we matched the training and testing noise, the reported variability pattern holds consistent across different levels of noise. (II) When RNNs were tested at lower noise than training, the reported variability pattern (Figure 5A) holds consistent and even stronger. For bias, there were attractive biases toward the prior or reward. (III) When RNNs were tested at higher noise than training, the oblique effect in variability broke down. Bias became more repulsive from prior or high-reward stimuli.

We further trained RNNs with varying levels of internal neural noise, where we randomly and uniformly selected an internal noise level for each of the 2000 epochs during training. After training, these RNNs were tested with a constant noise levels (Figure 12B). The variability amplitude was optimized for both prior and reward policy, with an increasing effect when tested with higher sensory noise setting. For bias, we also observed an amplified effect from prior and reward with higher testing noise. Compared to the RNNs trained with a constant level of internal noise, these RNNs trained with varying noise showed more robust effect of prior and bias, preserving the patterns regardless of the testing noise.

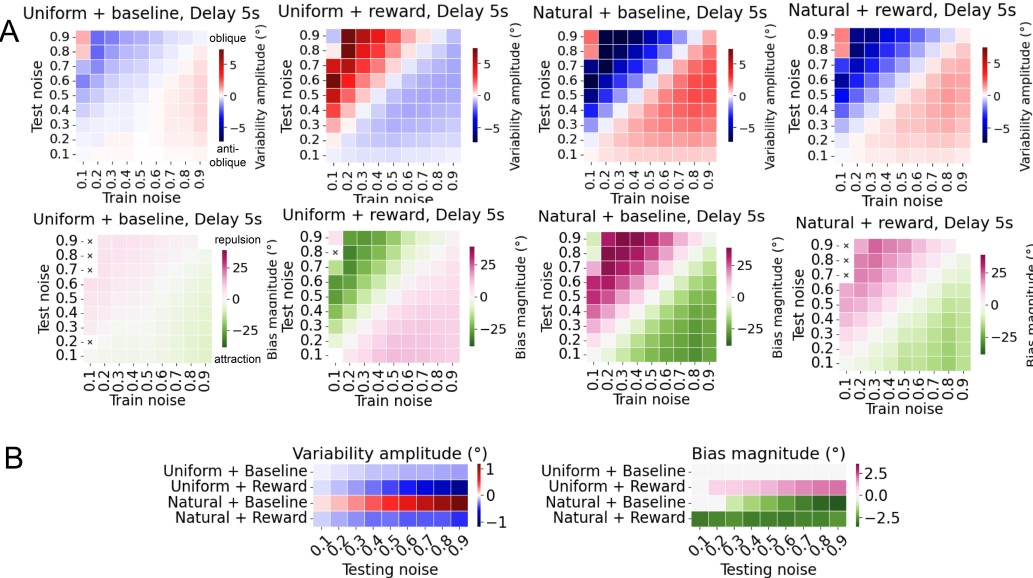

Figure 12: The effect of training and testing internal noise in RNNs WM. (A) RNNs variability amplitude and bias magnitude under different training and testing of internal noise level. For the variability, the red colors indicate an oblique (higher variability at diagonal orientations) while the blue colors indicate a pattern opposite to the oblique effect. For the bias, green colors indicate an attraction to the cardinal orientations while the pink colors indicate a repulsion. (B) The variability amplitude and bias magnitude for the RNNs trained with varying internal noise and tested with different level of internal noises.

