# OpenReview forum: "Efficient Allocation of Working Memory Resource for Utility Maximization in Humans and Recurrent Neural Networks"
_NeurIPS.cc/2025/Conference — NeurIPS 2025 poster_

### Official Review · Reviewer_QE77 · 2025-06-26

**Clarity:** 1
**Significance:** 2
**Originality:** 2
**Rating:** 3
**Confidence:** 3

**Summary:**

The authors study human behavior from a computational perspective in a delayed-estimation WM task. More specifically, they investigate how learned reward structures effect WM. No structure in learned rewards led to a sensitivity to naturalistic (in this case cardinal orientations) priors. This effect, however, could be overwritten based on learned rewards and was modulated by the delay of the estimation. This was then captured by ideal observer and recurrent neural network models.

**Questions:**

1. The authors write that “We decomposed memory errors into bias and bias-corrected (with a polynomial regression) variability.” It is not clear to me at all what was done here, but this seems particularly important as it is the foundation for the main analysis. A better explanation is desperately needed. I was also wondering why the composition was done in the first place. Why not look just at MSE or similar? That seems much simpler.
2. Do the authors think that, if the reward structure was communicated directly instead of learned, the effect would disappear? The discussion seems to hint at this at least. If so, where is this accounted for in the computational theory?
3. My guess is that the RNN is only modeling the WM component of the task, is that correct? Perhaps be good to mention that somewhere.
4. For the RNN models, the fact that they were trained with gradient descent does not really seem relevant. Wouldn’t you have gotten the same behavioral characteristics with any reasonable learning algorithm?

**Ethical Concerns:**

["NO or VERY MINOR ethics concerns only"]

**Final Justification:**

The authors' have addressed some of my concerns in the rebuttal, leading me to increase my score to 3. However, to really judge with confidence how the proposed changes would effect the readabiltiy of the paper, I would have to see how they are integrated in the text (which I don't think is easy to realize in this rebuttal-style format that doesn't allow uploading an updated PDF).

**Limitations:**

Discussion was sufficient.

**Paper Formatting Concerns:**

-

**Quality:**

2

**Strengths And Weaknesses:**

Strengths:
1. Introduction set up the work nicely and placed it into context.
2. The description of the behavioral task was easy to follow.
3. Figuring out how learned rewards interact with working memory is an important research direction and the results presented in Figure 3 were quite clear and show an interesting effect.

Weaknesses:
1. Looking at Figure 2, the only result that seemed clear to me was the oblique effect in Figure 2A. For instance, the authors write that, for the reward context, “diagonal orientations (now associated with higher reward) were maintained well while cardinal orientations (now associated with lower reward) had their memory precision dropped.” But that is not apparent to me looking at either Figure 2B or 2D.
2. I had a hard time following much of the modeling (even though I consider myself an expert in both ideal observer models and recurrent neural networks). In the present state, I don’t think it is ready for publication. Just to give some pointers:
* I_0 and I_1 on p4 were never introduced, making it impossible to understand what J is.
* The same equation is shown twice on p4.
* “We constructed the resource allocation as a function of orientations by fitting the model to individual data in different contexts and delays.” How this is done is not described at all.
* The first two equations on p5 contain an expectation but this is dropped for the third equation. The reason for this is unclear to me. There also seems to be a minus missing in the third equation.
* v(\theta) is never introduced. Is it the same as r(\theta)?
* J(\theta, t) is introduced but an expression is never given.

---

> ### Author Rebuttal · Authors · 2025-07-31
>
> We thank the reviewers for their constructive comments. We appreciate the reviewer’s positive comments highlighting the clear motivation and contextual framing, the well-explained behavioral task, and the importance and clarity of the work.
> To address issues raised, in the revision, we will include additional visualizations to clarify the results on reward context and the memory delay. We will add detailed explanations that are needed for the equations, symbols and how we decompose the bias and variability. Below we addressed each comment by the reviewer.
>
> **1. Clarify the behavioral effects in Fig. 2.**
>
> To address this point, we will add new figures and include more details of statistical results in the revision. Please see more details in our response point 1 to reviewer jK1W.
>
> **2. Clarify the modeling part.**
>
> Thank you for your helpful feedback regarding clarity in the modeling section. We apologize for the confusion and have taken steps to improve the manuscript accordingly.
> - **Define symbols.**
>   - We now explicitly define all mathematical symbols upon their first appearance to enhance readability and reproducibility, please see response point 5 to reviewer jK1W regarding the symbols.
>   - In addition, $J(\theta, t)$ in Sec 3.3 is the allocation of resource $J$ that is a function of orientation $\theta$ and time $t$. It is a direct extension of the equation $L = \int p(\theta)\ J(\theta)^{-\frac{p}{2}}\ d\theta $ at line 128, but here we introduce a time domain, allowing the resource $J$ to change with time. At Line 155, we found that even after considering the time domain, the $J_{opt}$ only depends on orientation. That is, the same strategy of resource allocation (more resource to more frequent stimuli) is the same across time points throughout memory delay. As a consequence, the difference in resources allocated between frequent (cardinal orientations) and infrequent (diagonal orientations) stimuli will accumulate over time, leading to a stronger oblique effect with longer delay, consistent with our observations in Fig. 2A.
> - **Repeated equation.** We will also correct the redundancy in the text where the same equation appeared twice on page 4 in the revised version.
> - **Clarify resource model fitting.** To break down the sentence “We constructed the resource allocation as a function of orientations by fitting the model to individual data in different contexts and delays.”:
>   - The memory error in each context and delay is separately fitted to the model
>   - We fitted a function $\bar{J}(\theta) = A \cdot \left(1 - B \cdot |\sin(2\theta + \phi)|\right)^n$ where the mean resource $\bar{J}$ is allowed to vary as a function of orientation $\theta$. The free parameters in the functions $A$, $B$, $\phi$ and $n$ are used so the function has the potential to flexibly form shapes such as a flat line or the “W” shape we found in the paper. We fitted this function using the memory errors (residuals) for each context and each delay separately.
>   - After fitting these free parameters to the memory error of each subject, context and delay, we visualize the distribution of resource $\bar{J}(\theta)$, by plotting the best-fit functions and averaged across subjects (Fig. 3)
>
> - **Correct the equation.** You are absolutely right that the expectation symbol was inadvertently dropped in the third equation.
>   - We will correct it in the revised manuscript to ensure consistency and mathematical accuracy: $ L = \int p(\theta)\ r(\theta)\ \mathbb{E}|\hat{\theta} - \theta| \ d\theta$.
>   - Note that the objective changes from maximizing the second equation (the reward) in Page 5 to minimizing the loss so there is no minus sign. To make it clearer, we change the left side of the third equation from $R$ (reward) to $L$ (loss).
>
> **3. Validate the bias-variability decomposition.**
>
> The memory error could come from both bias and variability as shown in Fig. 1G. For each context-delay pair, we performed a polynomial regression for the behavioral error as a function of stimulus orientation.
> - Specifically, we fit a low-degree polynomial (degree 3) regression to estimate the expected bias as a function of orientation.
> - The fitted bias $\hat{b}(\theta)$ was then subtracted from the raw errors to obtain bias-corrected errors (residuals) for subsequent variability analyses.
> - The variability (Fig. 2A and 2B) was computed as the standard deviations of these residuals.
> - The bias was computed by simply averaging the raw (signed) errors within each bins (Fig. 2C and 2D).
>
> In the revision, we will include a figure showing the bias and the fitted bias (from the polynomial regression).
>
> As we explained in Fig. 1G, estimation errors have two different components—variability and bias, and in the literature researchers have observed distinct patterns between them, especially for orientation estimation. For example, Girshick et al., 2011 separately reported variability and bias, and showed that the variability patterns can be linked to environmental prior. More recent work on efficient coding theory has attempted to explain the sign and magnitude of bias and variability (Wei & Stocker, 2015, 2017). Recent studies on human performance in orientation estimation also used the same approach to decompose variability and bias (Van Bergen et al., 2015). We believe that decomposing them is the most thorough way to look at the memory dynamics on orientations.
>
> **References**
>
> Van Bergen, R. S., Ji Ma, W., Pratte, M. S., & Jehee, J. F. (2015). Sensory uncertainty decoded from visual cortex predicts behavior. Nature neuroscience, 18(12), 1728-1730.
>
> Geurts, L. S., Cooke, J. R., van Bergen, R. S., & Jehee, J. F. (2022). Subjective confidence reflects representation of Bayesian probability in cortex. Nature human behaviour, 6(2), 294-305.
>
> **4. Direct instruction versus self-learned reward structure.**
>
> Indeed, we intentionally designed the reward manipulation to be learned through experience rather than directly instructed, though direct instruction will save much effort.
>
> Our rationale was grounded in both theoretical and empirical considerations. First, there is evidence from the literature that **learned reward associations—especially via reinforcement—can produce stronger and more flexible modulations of behavior** compared to explicit instructions (e.g., Collins & Frank, 2013; Ballard & McClure, 2019). Learning the reward contingencies engages brain systems involved in reinforcement learning and may promote **deeper integration of value information** into cognitive control and working memory mechanisms, potentially enhancing generalization or transfer.
>
> Second, in prior studies where reward structures were **explicitly cued or instructed**, effects on working memory resource allocation have often been absent (Brissenden et al., 2023, and Van den Berg et al., 2023). We hypothesized that **allowing participants to learn the value structure through trial and error would produce stronger reward-dependent modulation of memory**, which aligns with the robust effects we observed.
>
> We agree this distinction between learned vs. instructed value could have theoretical importance, and we plan to explore this manipulation more directly in future work.
>
> **5. RNN is only modeling the WM component of the task.**
>
> **Yes, that is correct.** The RNN was designed to model the working memory (WM) component of the task, rather than the initial learning of the reward-orientation mapping in the value-learning task. The RNN model itself was trained to optimize performance in the WM task, capturing the influence of the learned value structure through reward-weighted loss during training, without explicitly modeling the value learning task in the behavioral experiment. We will clarify this point more clearly in the revised manuscript.
>
> **6. Training algorithm of the RNN**
>
> We used backpropagation through time (BPTT) and stochastic gradient descent (SGD) for training, but we do not think that our results hinge on the specific training algorithm. We recognize that developing biologically plausible alternatives to BPTT remains an active area of research, and several notable approaches have demonstrated similar outcomes (e.g., Miconi, 2017; Murray, 2019; Bellec et al., 2020). We will revise the Discussion to acknowledge this point and clarify that our findings likely reflect properties of the trained model architecture and task objective, rather than being specific to the training algorithm used.
>
> **References**
>
> Miconi, T. (2017). Biologically plausible learning in recurrent neural networks reproduces neural dynamics observed during cognitive tasks. Elife, 6, e20899.
>
> Murray, J. M. (2019). Local online learning in recurrent networks with random feedback. Elife, 8, e43299.
>
> Bellec, G., Scherr, F., Subramoney, A., Hajek, E., Salaj, D., Legenstein, R., & Maass, W. (2020). A solution to the learning dilemma for recurrent networks of spiking neurons. Nature communications, 11(1), 3625.

---

> > ### Comment · Reviewer_QE77 · 2025-08-01
> >
> > Thanks for the response, based on which I'll increase my score to 3 but with some uncertainty. To really judge with confidence how the proposed changes would effect the readabiltiy of the paper, I would have to see how they are integrated in the text (which I don't think is easy to realize in this rebuttal-style format that doesn't allow uploading an updated PDF).

---

> > > ### Author Response · Authors · 2025-08-01
> > > **Thank you**
> > >
> > > Thank you so much for your thoughtful follow-up and for considering increasing the score. We completely understand that it’s difficult to fully assess the readability without being able to view an integrated pdf version. In the camera-ready version, if accepted, we will take great care to incorporate all the suggested improvements, with particular attention to enhancing the clarity of the paper. We're grateful for your constructive feedback and comments, which have helped improve our work.

---

> > > ### Author Response · Authors · 2025-08-07
> > > **Follow-up on Score Update**
> > >
> > > Thank you again for your thoughtful feedback on our submission. We noticed you mentioned in your comments that you might increase the score to 3, but the score in the system still shows as 2. We just wanted to check in case there was any technical issue or if the update didn’t go through from the review form.

---

### Official Review · Reviewer_jK1W · 2025-06-30

**Clarity:** 2
**Significance:** 3
**Originality:** 3
**Rating:** 4
**Confidence:** 4

**Summary:**

The manuscript investigates how precision in human working memory (WM) is allocated by both environmental statistics and learned reward contingencies with limited memory resources. The authors derive an efficient-coding rule in which Fisher information is proportional to the product of the stimulus distribution and a value term, and they show that a recurrent neural network trained under corresponding loss functions reproduces the human patterns. Together, these results support that WM precision is dynamically steered toward stimuli with higher prior and values.

**Questions:**

1.	Could you report formal comparisons of variability parameters between baseline and reward contexts?

2.	Please define all symbols in formulas.

3.	Have you trained RNNs without the activity regularized, or with a weaker coefficient? If the reward-dependent allocation persists, that would suggest that resource constraint is not an explanation.

4.	Can you justify the power of sample size?

**Ethical Concerns:**

["NO or VERY MINOR ethics concerns only"]

**Final Justification:**

My concerns are adequately addressed.

**Limitations:**

No, there’s no explicit limitations mentioned in the discussion.

**Quality:**

3

**Strengths And Weaknesses:**

Strengths

•	The study combines behavioral data, a normative model, and RNN simulations, providing a multi-angle examination of the hypothesis. The motivation is clear.

Weaknesses

•	In Fig. 2 the difference between baseline and reward contexts is not obvious by eye; no statistical comparison of fitted key parameters (e.g., A,B, phi,n) is reported.

•	Symbols (e.g., epsilon, kappa, I1, I0) between line 99 and 104, in Eq. 1 and  (line 142) v(theta) are never defined, which hampers reproducibility.

•	Line 155 claims that the J_opt depends on t; but the explicit formula is unclear.

•	The RNN uses an L2 activity penalty. It is not shown whether removing or relaxing this penalty changes the results, leaving open whether “resource limits” drive the effect.

•	Fourteen participants limit generalizability.

---

> ### Author Rebuttal · Authors · 2025-07-31
>
> We thank the reviewers for their constructive comments, and especially for recognizing our approach that integrates new behavioral data, cognitive modeling, and neural network models, providing multi-angle examinations on the research topic. To address the issues raised, in the revision, we will describe more statistical tests and include additional visualization to clarify the effect or reward in behavioral results. We will also add explanations on symbols in equations, which were missing in the original manuscript. Below we addressed each comment by the reviewer.
>
> **1. Formal comparisons of variability between baseline and reward contexts**
>
> For better visualization and statistical analysis of the **behavioral results** in Fig. 2, we will do the following:
>
> - (1) add a figure which doesn’t use orientation as the x-axis, but uses the “distance to the cardinal orientations” as the axis (bin1 to bin5, from the most cardinal to the most diagonal orientations; this involves folding the original x-axis in Fig. 2 twice). The bonferroni corrected paired t-test on this type of data binning showed that:
>   - For baseline context, the variability in delay=5s is significantly higher compared to the delay=1s in the bin4 (relatively close to diagonal, *p* < .01).
>   - For reward context, the variability in delay=5s is significantly higher compared to the delay=1s in the bin2 (relatively close to the cardinal, *p* < .05)
> - (2) add figures which change the way we group the results: in each delay condition, we will plot the variability and bias in reward and baseline context to make the comparison more obvious. Since there was a significant 3-way interaction in variability indicated by a linear mixed-effects model (context x delay x orientation, *p* < .05), we also provide the statistical results grouped by the delays.
>   - For delay=1s, there is no significant 2-way interaction (context x orientation, *p* = .161), indicating the pattern of the variability over orientation remain similar across the contexts;
>   - For delay=5s, there is a significant 2-way interaction (context x orientation, *p* < .05), showing that the pattern of the variability over orientation is significantly different across two contexts.
>
> Additionally, to address this question, For the **modeled resource distribution** in Fig. 3, we will also add a new figure that groups the resource distribution by delay in the revised version. Based on the cluster-based permutation test, there is no significant different orientation cluster for resource across contexts in delay=1s, while two diagonal clusters were identified for delay=5s where there are more resource allocated to the diagonal (high-reward) stimuli in the reward context compared to the baseline (*p* < .001).
>
> **2. Statistical comparison of fitted parameters.**
>
> For the formula $\bar{J}(\theta) = A \cdot \left(1 - B \cdot |\sin(2\theta + \phi)|\right)^n$, Fig.3 shows that resource distributions exhibited **a stronger oblique effect in long delay in the baseline context** (the “W” pattern in Fig.3A), but **a weaker oblique effect in long delay in the reward context** (a shallower “W” in Fig 3B). These effects are **reflected in the parameter $B$**. Indeed, we found a significant interaction effect (context x delay, *p* < .05) when testing the best-fitted $B$ values. We will report the statistical tests on all the fitted parameters in the revision. We describe them in more details here:
> - For $A$: the linear mixed-effects model showed no significant interaction effect (context x delay, *p* > .05), and no significant main effect (*p* > .5). The bonferroni corrected one-sample t-test showed that A is significantly larger than 0 (*p* < .001) in all contexts and delays.
> - For $B$: the linear mixed-effects model showed a significant interaction effect (context x delay, *p* < .05), and a marginal significant effect for delay (*p* = .06). The bonferroni corrected one-sample t-test showed that B is significantly larger than 0 (*p* < .001) in all contexts and delays.
> - For $\phi$: the linear mixed-effects model showed a marginal interaction effect (context x delay, *p* = .06), and no significant main effect (*p* > .05). The bonferroni corrected one-sample t-test showed that $\phi$ is only significantly larger than 0 in delay=5s in both context, while not significantly different from 0 in delay=1s in both context.
> - For $n$: the linear mixed-effects model showed no significant interaction effect (context x delay, *p* > .05), but a significant effect for delay (*p* < .05). The bonferroni corrected one-sample t-test showed that n is significantly larger than 0 (*p* < .001) in all contexts and delays.
>
> **3. Activity regularization and working memory resources.**
>
> In addition to the models reported in the main paper, we also trained RNNs without weight or activation regularization while keeping all the other settings the same. We found similar behavioral patterns even without the regularization. This suggested that resource-efficient behavior can emerge from task optimization alone, even in the absence of hard-coded limitations. See details in response point 3 to reviewer Y8u3 regarding "Alternative loss weightings".
>
> **4. Sample size and generalizability.**
>
> At this stage, we are not able to provide additional data or replication, but we did conduct a power analysis regarding the number of subjects. The result showed that the 14 subjects achieved well above 90% power to detect an effect of reward in the oblique orientation. See response point 1 to reviewer Y8u3. Additionally, our study uses a tightly controlled cognitive task with a within-subject design that is typical in cognitive experiments.
>
> **5. Define all symbols in formulas.**
>
> Thank you for pointing out the missing symbol definitions. We now explicitly define all mathematical symbols upon their first appearance for clarity and reproducibility, and they will be added to the revised manuscript near their respective equations.
> - Specifically on **Page 4**:
>   - For von Mises distribution equation, $\epsilon$ denotes the angular memory error, and $\kappa$ is the concentration parameter.
>   - $I_0(\kappa)$ and $I_1(\kappa)$ are the modified Bessel functions of the first kind of order 0 and 1, respectively.
>   - $J \sim \mathrm{Gamma}(\bar{J}/\tau, \tau)$ specifies that memory precision varies across trials, drawn from a Gamma distribution with shape parameter $\bar{J}/\tau$ and scale parameter $\tau$.
>     - $\bar{J}$ is the mean of the resource parameter $J$.
>     - $\tau$ is the scale parameter of the Gamma distribution, controlling variability in $J$ across trials.
> - On **page 5**:
>   - $v(\theta)$ in line 142 is a typo which should be context-dependent rewards $r(\theta)$, and $\theta$ is the orientation.
>   - $C$ in the equation $\int J(\theta)^{\beta} \ d\theta < C $ represents total resource constraint, and $\beta$  is an exponent term used in the power-law efficient code (Morais and Pillow, 2018)
>   - $p$ from $L = \int p(\theta)\ J(\theta)^{-\frac{p}{2}}\ d\theta $, and $q$ from $ J_{\text{opt}} \propto p(\theta)^q \$ are also exponent terms used in the power-law efficient code (Morais and Pillow, 2018)
>
>
> **6. Clarify Line 155 $J_{opt}$ depends on $t$.**
>
> In section 3.3 we introduced $J(\theta, t)$, which is the allocation of resource $J$ that is a function of orientation $\theta$ and time $t$. It is a direct extension of the equation $L = \int p(\theta)\ J(\theta)^{-\frac{p}{2}}\ d\theta $ at line 128, but here we introduce a time domain, allowing the resource $J$ to change with time.
>
> The results stated in Line 155 $J_{opt}(\theta,t) \propto p(\theta)^q$ is the outcomes of optimization (minimizing the loss) in Line 154. We found that even after considering the time domain, the $J_{opt}$ only depends on orientation. That is, the same strategy of resource allocation (more resource to more frequent stimuli) is the same at any time points throughout the memory delay. As a consequence, the difference in resources allocated between frequent (cardinal orientations) and infrequent (diagonal orientations) stimuli will accumulate over time, leading to a stronger oblique effect with longer delay (consistent with our data Fig. 2A).

---

> > ### Comment · Reviewer_jK1W · 2025-08-05
> >
> > Thank you for your thorough responses. I consider my concerns to be adequately addressed. I will update my score.

---

> ### Author Response · Authors · 2025-08-06
> **Thank you**
>
> Thank you so much for your thoughtful feedback and for updating your score. We're glad that our responses addressed your concerns, and we appreciate your comments—they’ve helped improving our paper.

---

### Official Review · Reviewer_Tu7B · 2025-07-02

**Clarity:** 3
**Significance:** 2
**Originality:** 3
**Rating:** 4
**Confidence:** 3

**Summary:**

The paper examines the question on the basis of what humans can effectively allocate working memory resources - utility or environmental statistics. Experiments with real participants and model experiments with RNN were conducted.

**Questions:**

1. How many people took part in the experiment?
2. How correct is it to use RNN as a memory model?
3. Are there other approaches to assessing the allocation of working memory resources?
4. It is useful to conduct experiments with a very simple model, MLP, to demonstrate that the memory in the RNN is actually used and the model does not simply remember the pattern present in the data.

At the moment my rating is “Borderline reject”, but I will consider increasing the rating if the answers to the questions are convincing.

**Ethical Concerns:**

["NO or VERY MINOR ethics concerns only"]

**Final Justification:**

The authors provided sufficient clarifications to my questions, which allows me to increase my score. However, since taking these answers into account in the final version of the paper requires its significant revision, which cannot be assessed, I will increase the score to 4.

**Limitations:**

The limitations of research are discussed sufficiently in the Discussion section, but I recommend that authors describe limitations into a separate section to improve the perception of the work.

**Paper Formatting Concerns:**

I did not notice any major formatting issues.

**Quality:**

3

**Strengths And Weaknesses:**

**Strengths:**
1. A comparison was made with the results of real participants.

**Weaknesses:**
1. A very simple model, RNN, is used for comparison. Its use for assessing the mechanisms of human memory raises questions.

**Quality**

The experiments sufficiently confirm the authors' сlaims on the issues discussed in the paper.

**Clarity**

The paper is well written and easy to read.

**Significance**

As far as I can judge from my limited expertise, the research presented in the paper is important.

**Originality**

The paper lacks a Related Work section and a brief survey of relevant work given in the Introduction section. As far as I can tell, the work is quite original, but I recommend that the authors expand the comparison section and put it into a separate section.

I should note that I am not an expert in neuroscience and cognitive science and may be missing important results for these areas and incorrectly evaluate the significance and originality of the work.

---

> ### Author Rebuttal · Authors · 2025-07-31
>
> We thank the reviewers for their constructive comments, and for recognizing that the research topic we address is important. Based on the reviewer’s comments, in the revision, we will expand the Introduction to justify how RNNs are widely used in neuroscience literature to study working memory, and we will add a dedicated Related Work section and Limitation section to improve the clarity. In addition, we showed that RNNs can generalize to untrained longer delays and untrained stimuli. Below we addressed each comment by the reviewer.
>
> **1. Validate the use of RNN for assessing the mechanisms of human memory.**
>
> We agree that RNNs are simplifications of biological memory systems. Our goal is not to claim that RNNs fully capture the mechanisms of working memory, but to use them as functional models to explore how reward and prior information shape memory dynamics.
>
> RNNs are widely used in neuroscience to model working memory due to their ability to maintain information over time and generate dynamics resembling cortical activity observed in humans and non-human primates (e.g., Compte et al., 2000; Wang, 1999; Bouchacourt & Buschman, 2019; Wimmer et al., 2014; Esnaola-Acebes et al., 2022; Yang & Wang, 2020; Wang, 2021). When trained on task-relevant objectives, they often develop ring-attractor-like dynamics, thought to underlie memory for simple visual features  in the brain.
>
> While RNNs have been extensively used to study working memory, their adaptation to varying reward structures and stimulus statistics remains underexplored. We deliberately used a minimal architecture to isolate the influence of these factors. Despite its simplicity, the model reproduced key behavioral patterns observed in human data. Overall, we used RNN as the model system for working memory based on the rich literature on this topic. In the revision, we will expand the background on how RNNs are used to study memory in neuroscience and clarify our rationale for using minimal models to study the influence of training context.
>
> **References**
>
> Wimmer, K., Nykamp, D. Q., Constantinidis, C., & Compte, A. (2014). Bump attractor dynamics in prefrontal cortex explains behavioral precision in spatial working memory. Nature neuroscience, 17(3), 431-439. doi: 10.1038/nn.3645.
>
> Compte, A., Brunel, N., Goldman-Rakic, P. S., & Wang, X. J. (2000). Synaptic mechanisms and network dynamics underlying spatial working memory in a cortical network model. Cerebral cortex, 10(9), 910-923. doi: 10.1093/cercor/10.9.910.
>
> Bouchacourt, F., & Buschman, T. J. (2019). A flexible model of working memory. Neuron, 103(1), 147-160. doi: 10.1016/j.neuron.2019.04.020
>
> Wang, X. J. (2021). 50 years of mnemonic persistent activity: quo vadis?. Trends in Neurosciences, 44(11), 888-902.
>
> Yang, G. R., & Wang, X. J. (2020). Artificial neural networks for neuroscientists: a primer. Neuron, 107(6), 1048-1070.
> Esnaola-Acebes, J. M., Roxin, A., & Wimmer, K. (2022). Flexible integration of continuous sensory evidence in perceptual estimation tasks. Proceedings of the National Academy of Sciences, 119(45), e2214441119.
>
> **2. Add Related Work section in Introduction**
>
> To clarify, in the original manuscript, we discuss **relevant work** in the Introduction and Section 3. These include:
> - Foundational findings on working memory (WM) capacity and resource limit (Luck & Vogel, 1997, 2013; Ma et al., 2014; Pertzov et al., 2017),
> - Evidence for flexible allocation of WM resource based on attention (Zhang & Luck, 2008; Yoo et al., 2018),
> - Contrasting non-effect evidence on reward-modulation of WM resource (Brissenden et al., 2023; Van den Berg et al., 2023),
> - Neural and computational models of WM using RNNs and attractor dynamics (Compte et al., 2000; Wimmer et al., 2014; Bouchacourt & Buschman, 2019), and
> - Normative models of efficient coding and utility optimization under constraints (Wei & Stocker, 2015; Morais & Pillow, 2018; Hahn & Wei, 2024).
>
> **Research gap:** While these studies have advanced our understanding of WM and its limitations, it remains unclear whether and how WM resources can be allocated efficiently to maximize utility. This gap spans both behavioral and modeling perspectives. Our work fills this gap by:
> - Providing behavioral evidence that WM resources are adaptively allocated to reward-maximizing features after value learning,
> - Proposing a framework that explains how prior and reward jointly influence WM resource allocation, and extending the model to incorporate temporal dynamics,
> - Using simple but interpretable RNN models to show that such adaptive allocation can emerge through training objectives and stimulus distributions.
>
> Based on this and the previous comment from the reviewer, In the revision, we will expand and consolidate a dedicated Related Work section to highlight the previous studies, especially the neuroscience studies that used RNN to model working memory and the research on the effect of reward on working memory, and the research gaps to better explain our contributions.
>
> **3. Sample size.**
>
> There were 14 participants who completed the full experiment (14 subjects X 2 sessions X 90 minutes per session), and we trained 15 RNNs in each context and prior distribution conditions. We also conducted a power analysis regarding the number of subjects. The result showed that the 14 subjects achieved well above 90% power to detect an effect of reward in the oblique orientation. See response point 1 to reviewer Y8u3.
>
> **4. Other assessment of the allocation of working memory resources.**
>
> Yes, there are several well-established approaches to studying working memory resources in cognitive science.
> - One line of research investigated the format and the capacity of working memory by studying how memory precision declines when subjects were asked to remember more items (e.g., Cowan, 2002; Zhang and Luck, 2008; van den Berg et al., 2012).
> - The other line of study focused on the effect of attentional cue on working memory. They involved asking subjects to remember multiple items in a trial and at the same time presenting an attentional cue to inform the subjects which item is more important (more likely to be probed/asked at the end). The general findings are that people remembered the cued item better than the un-cued items. Note that this kind of attentional cueing effects are well-established (cited studies in the introduction: Zhang and Luck, 2008, Dube et al., 2017, Yoo et al., 2018) and are not the research interest in our study. Here we aim to isolate the effect of reward by implementing a working memory task with a single item per trial.
>
> Besides the manipulation of reward in the behavioral experiment, we also implemented a L2 regularization on total neural activity in the RNN, which can be considered as a form of resource constraints (see more in the response to Reviewer jY8u3’s point 3).
>
> **References**
>
> N Cowan, The magical number 4 in short-term memory: A reconsideration of mental storage capacity. Behav Brain Sci 24, 87–114, discussion 114–185. (2001).
>
>  Zhang, W., & Luck, S. J. (2008). Discrete fixed-resolution representations in visual working memory. Nature, 453(7192), 233-235.
> Van den Berg, R., Shin, H., Chou, W. C., George, R., & Ma, W. J. (2012). Variability in encoding precision accounts for visual short-term memory limitations. Proceedings of the National Academy of Sciences, 109(22), 8780-8785.
>
> **5. RNNs do not simply remember the input-output patterns.**
>
> Thanks for raising an important point about ensuring that the RNN uses its memory dynamics rather than memorizing input-output patterns. To address this, we evaluated the memory in (1) **longer delays** than it was trained on and (2) **unseen stimuli**.
> - Even at delays of 6–8s (trained with ~4s), the RNN maintained the memory well, with a low memory variability (<8°). We also observed robust effects of prior and reward on memory variability, consistent with adaptive use of memory over time.
> - We trained and tested RNNs in different levels of recurrent weight noise levels (σ = 0.1, 0.5, 0.8), and observed the reported variability pattern (Fig. 5) holds consistent when the models are tested in lower noise than they are trained. Further details are discussed in response point 3 to the reviewer Y8u3.
>
> Please also see our responses point 5 to reviewer Y8u3, where we now show that the **RNN recurrent weights are needed to explain the data**. Changing the readout weights alone can not reproduce how memory variability changes with time, prior and reward policy.
>
> We agree that simpler models like MLPs can be useful baselines. We trained an MLP on the same task and found that it could accurately classify orientations when evaluated immediately after stimulus offset. Previous studies also found that DCNN and MLP trained on natural images showed higher sensitivity to cardinal orientations which are more prevalent in natural environments, similar to humans and our results here (the cited study in our manuscript: Benjamin et al., 2022). However, these models are purely feedforward without temporal dynamics, and therefore are not well-suited for modeling memory dynamics, which is the focus here. In contrast, RNNs have a dynamic internal state that evolves without additional input.
>
> That said, if the reviewer has a specific use in mind for how an MLP could help elucidate the issue further, we would be very happy to add that analysis in future work.
>
> **References**
>
> Benjamin, A. S., Zhang, L. Q., Qiu, C., Stocker, A. A., & Kording, K. P. (2022). Efficient neural codes naturally emerge through gradient descent learning. Nature Communications, 13(1), 7972.
>
> **6. Describe limitations into a separate section**
>
> Thank you for the helpful suggestion. We agree that clearly stating the limitations in a dedicated section can improve clarity and transparency and we will do so in the revised manuscript.

---

> > ### Comment · Reviewer_Tu7B · 2025-08-08
> > **Answer to the authors**
> >
> > Thanks to the authors for the clarification. Taking them into account I will increase my score.

---

> > > ### Author Response · Authors · 2025-08-08
> > > **thank you**
> > >
> > > Thank you for considering our clarifications and for updating your score — we truly appreciate your thoughtful feedback.

---

> ### Comment · Area_Chair_nW97 · 2025-08-05
> **Please respond to the rebuttal**
>
> Dear Reviewer Tu7B
>
> Given that the Discussion period will end in 48 hours, could you please read the authors' rebuttal and provide your comment? Thanks.
>
>
> Best
> AC

---

### Official Review · Reviewer_Y8u3 · 2025-07-03

**Clarity:** 3
**Significance:** 2
**Originality:** 3
**Rating:** 4
**Confidence:** 3

**Summary:**

The manuscript investigates whether humans can efficiently allocate their working memory (WM) resources to maximize expected reward, and whether RNNs trained under similar constraints exhibit the same behavior.

Behavioral side: 14 adult participants first learned stimulus–reward mappings and then performed an orientation-recall task after 1s or 5s delays. When rewards were uniform, memory precision followed the classical oblique effect (better for cardinals). When high rewards were assigned to oblique orientations, that pattern gradually inverted over the delay, indicating value-guided WM maintenance.

Computational side: RNNs (32 units) trained with BPTT yield results that resemble those from human behavioral data.

**Questions:**

The paper assumes participants have fully learnt the reward schedule after about 100 trials. However, the asymptotic performance is only shown descriptively. Could you provide model-based learning-curve fits and exclude incomplete learners?

Training via BPTT is hard to accomplish in a biological system. Can you obtain similar results with frozen recurrent weights or without unfolding the computational graph in time (or some other non-BPTT training)?

**Ethical Concerns:**

["NO or VERY MINOR ethics concerns only"]

**Final Justification:**

Solid work. Authors have addressed my comments, although (as another reviewer pointed out) it is a bit difficult to judge all the updates in this format (without seeing detailed results implemented in pdf).

**Limitations:**

yes

**Paper Formatting Concerns:**

No issues.

**Quality:**

2

**Strengths And Weaknesses:**

Strengths:

Whether WM reflects learned value, not just attention, is still debated; the paper provides relevant behavioural evidence as well as a modeling framework.

Experimental design is clever. By making reward oppose the natural orientation prior, the design neatly teases apart prior vs. value effects.

Combining data, theory and modeling with RNNs helps triangulate the mechanism and increases the work’s breadth.

Variability–bias decomposition, mixed-effects modelling and VP fits are appropriate and clearly reported.


Weaknesses:

Small sample size (14 subjects) limits statistical power.

Reward effects appear only in variability, not bias. The discussion speculates about encoding vs. maintenance, but the normative framework itself ignores bias.

Only one tiny architecture is reported. It is unclear whether the qualitative results survive with larger networks, different noise levels, or alternative loss weightings.

---

> ### Author Rebuttal · Authors · 2025-07-31
>
> We thank the reviewers for their insightful comments. We are especially encouraged by the recognition of our study’s key strengths, including the clever experimental design and the integration of behavior and modeling. In response to the feedback, we conducted a power analysis, expanded the discussion on bias, and performed systematic tests examining the role of RNN architecture, noise levels, loss weightings, and transfer learning. We believe that these additions have strengthened our manuscript. Below, we address each point in detail.
>
> **1. Statistical power and sample size**
>
> There were 14 participants who completed the full experiment, and we trained 15 RNNs in each condition.
> We conducted power analysis based on effect sizes reported in a prior study (Exp. 1 in Schaffner et al., 2023, which investigated the effect of reward on orientation perception). Specifically, we used the reported coefficient (Estimate = −0.16, SE = 0.09) from the Bayesian hierarchical linear regression model to approximate a standardized effect size (Cohen’s f ≈ 2.0). The effect reported estimates how reward influence the performance in oblique orientations. Since our design includes 5 orientation bins, we applied a Bonferroni correction for multiple comparisons, resulting in a corrected α = 0.01. Power analysis revealed that with 14 participants, our study already achieves well above 90% power to detect an effect of reward in the oblique orientation.
>
> **2. Include a discussion of the bias predicted by the normative model.**
>
> Thank you for the insightful comment. Indeed, our analysis, theoretical framework, and RNNs focused on changes in variability over time, rather than jointly explaining bias and variability. While we briefly acknowledged this in the discussion, we agree it deserved expansions in discussion.
>
> Efficient coding theory has successfully jointly explained perceptual bias and discrimination thresholds (Wei & Stocker, 2015, 2017). A direct extension of this framework to memory would predict that **bias and variability remain coupled across delays**. However, our results diverge from this: while variability increases substantially over longer delays, bias remains relatively stable. At short delays, the observed bias and variability align with prior empirical findings and theoretical predictions, suggesting that performance reflects established principles of perceptual encoding. The **temporal dissociation between bias and variability** highlights a distinction between encoding and memory maintenance—a key contribution of our study.
>
> We see two possible explanations:
> - (1) Recent theoretical work (e.g., Hahn & Wei, 2024) showed that bias reflects the combination of two opposing forces: a repulsive component from encoding (asymmetric likelihood function) and an attractive component from the prior used during Bayesian decoding. These effects can co-exist and vary independently. If, across delay or conditions, repulsion increases while prior-driven attraction also grows, their net influence may cancel, resulting in stable observed bias.
> - (2) WM maintenance engages neural populations distinct from those involved in initial sensory encoding. While encoding likely relies on sensory areas with fixed tuning curves that tile the feature space—as assumed in efficient coding models—WM relies on distributed activity in prefrontal and parietal cortices, where neurons show mixed and flexible selectivity. These dynamics may not preserve the original encoding structure, allowing variability to increase over time without corresponding changes in bias. This pattern aligns with our RNN simulations, which better capture memory-related dynamics than static sensory encoding.
>
> Based on another reviewer’s comment, we investigated more systematically, regarding how noise levels affect memory in RNNs. When noise levels are the same between training and testing, we found that bias is relatively stable compared to variability (consistent with our data). Interestingly when varying noise from training and testing we could observe stronger effect of delay on the bias (see response to the point 3)
>
> **3. Results from larger networks, different noise levels, or alternative loss weightings.**
>
> **Larger network.** We trained larger RNNs with 128 recurrent units (vs. 32 in the main paper). The results of variability (Fig. 5) are similar under a larger network, while the absolute variance was lower, suggesting improved precision with larger networks. Interestingly, bias patterns diverged: larger RNNs showed stronger biases over time—specifically, increased attraction to the prior in the baseline context, and to reward in the reward context. These patterns are not consistent with the repulsion we and others found for orientations in WM.
>
> **Noise levels.** For the smaller RNN originally reported in our manuscript, we systematically investigated RNNs trained and tested under different levels of internal sensory noise (σ = 0.1, 0.5, 0.8 of the recurrent weight). Our key findings are:
> - Matched Training and Testing Noise:
>   - The reported variability pattern (Fig. 5) holds consistent across different levels of noise.
>   - For bias, there is a repulsion to the prior in low noise (σ = 0.1) but transitioned into attraction when noise increased to σ = 0.8.
> - Testing at *lower* noise than training (e.g. trained on σ = 0.5, tested on σ = 0.1).
>   - The reported variability pattern (Fig. 5) holds consistent.
>   - There were attractive biases toward the prior or reward dominate, when tested with lower noise.
> - Testing at *higher* noise than training (e.g. trained on σ = 0.1, tested on σ = 0.5)
>   - The variability results we reported in the paper broke down.
>   - Bias became more repulsive from prior or high-reward stimuli, consistent with larger noise amplifying repulsive bias.
>
> **Alternative loss weightings.** we trained additional RNNs with zero L2 penalty on recurrent weights and activations, using zero initialization to prevent overfitting. We observed that, compared to the regularized models (1e-4 L2 penalty):
> - These models exhibited behavior holds consistent, whit variability affected by both prior and reward.
> - They developed similar recurrent weights (local excitation, broad inhibition) consistent with ring-attractor dynamics.
> - These results suggest that resource-efficient behavior can emerge from task optimization alone, even in the absence of hard-coded limitations. Thus, “limited resources” in working memory may reflect rational strategy to task goals rather than strict architectural bottlenecks.
>
> **Conclusions.** Across these settings, our core findings—variance modulation by prior and reward, and the emergence of attractor-like dynamics—remain qualitatively stable. Including these results in the revision will significantly improve the manuscript.
>
> **4. Learning results across 100 trials.**
>
> We agree that quantifying learning trajectories would provide a more rigorous picture of reward learning. In both the reward and baseline context, the participants completed 5 runs of learning, with 20 trials per run 100 trials in total. In the reward context, the percentage of trial participants choosing the diagonal (highest-reward) stimuli is already at 56% in the first run (chance level = 25%). This percentage increased over time and plateaued in the third run close to 70% and remained at the same level throughout the training. When binning the trials into smaller bins (5 trials per bin), we found that learning reached an asymptote (70% choosing the highest-reward stimuli) at around 60 trials. These patterns were consistent across almost all participants.
>
> In the baseline context, the choices for the diagonal stimuli (now not high-reward) of all participants reaches the chance level 25% around trial 65 which is at run 4, with all the participants near 25% in the final block. In the revised manuscript, we will add figures to illustrate these learning curves.
>
> **5. Results from frozen recurrent weights.**
>
> Thanks for this excellent and important question! We performed two sets of transfer learning experiments:
> - **Transfer across prior distributions (uniform → natural).** We first trained the 15 RNNs under a uniform distribution + baseline context. After training, we froze the recurrent weights and **re-trained only the readout weights** in the natural distribution + baseline context. We found that:
>   - The variability pattern remained similar to the original uniform + baseline models: variability was uniform across orientations and did not reflect the natural distribution prior.
>   - In contrast, the bias showed a shift toward the natural distribution + baseline pattern.
> - **Transfer across contexts (baseline → reward).** We trained 15 RNNs in the uniform distribution + baseline context, then froze the recurrent weights and retrained the readout in the uniform distribution + reward context. We found:
>   - For variability, there was a reduction at high-reward orientations, but this effect was weaker than in models fully trained in the reward context. Hence, both recurrent and readout weights contribute to reward-driven variability modulation.
>   - For bias, the transferred models closely matched the fully trained reward context models.
>
> Therefore, the effect of prior or reward policy on memory **variability is rooted in the changes of recurrent dynamics**, while **the bias can be more flexibly shaped by the readout layer based on prior and reward**.
>
> **Training algorithm.** We used BPTT and SGD for training, but we do not think that our results hinge on the specific training algorithm. We recognize that developing biologically plausible alternatives to BPTT remains an active area of research, and several notable approaches have demonstrated similar outcomes (Miconi, 2017, Murray, 2019, and Bellec et al., 2020). Nonetheless, BPTT and SGD remain the most widely used training methods for RNNs today. We will acknowledge this point in the revised Discussion section.

---

> > ### Comment · Reviewer_Y8u3 · 2025-08-06
> >
> > Thank you for your detailed responses. The additional analyses help strengthen the result. It would be good to ensure that you quantify all the observed effects and evaluate their significance. For instance, for your observation with frozen weights: "weaker than in models fully trained in the reward context", it's not clear whether the difference is significant.

---

> > > ### Author Response · Authors · 2025-08-07
> > > **thank you**
> > >
> > > Thank you very much for your thoughtful feedback and for updating your score. We really appreciate your suggestions throughout the review process — they’ve helped us improve the paper.
> > >
> > > In the revised version, we will formally quantify the observed effects by fitting raised-sinusoidal functions to the variability, reporting the maximum attraction or repulsion to summarize the bias, and conducting statistical analyses on these indices (e.g., the amplitude of the fitted sine functions). We believe these additions will further strengthen our results.

---

> ### Comment · Area_Chair_nW97 · 2025-08-05
> **Please respond to the rebuttal**
>
> Dear Reviewer Y8u3
>
> Given that the Discussion period will end in 48 hours, could you please read the authors' rebuttal and provide your comment? Thanks.
>
>
> Best
> AC

---

### Note · Authors · 2025-08-13

We thank all reviewers and the AC for their time, feedback, and engagement, which have led to substantial improvements to our paper. During the rebuttal phase, we:

**1. Addressed statistical power and key behavioral findings**

- Reported power analysis: our sample size has >90% power to detect reward effects on orientation estimation.
- Highlighted the temporal dissociation of bias and variability as a novel finding and outlined mechanistic explanations, by integrating recent theoretical frameworks.

**2. Strengthened RNN analysis through systematic testing and controls**

- Performed systematic testing of our RNN models by varying network size, noise levels during training/testing, and regularizations. These analyses showed that the main results—the effects of reward and prior on memory variability, and attractor-like dynamics—are robust, while revealing conditions where the bias of RNNs aligned with human behavior.
- Showed that RNNs generalize to (1) delays longer than those used in training and (2) unseen stimuli, while maintaining robust prior/reward effects, confirming their reliance on dynamic memory rather than static input–output mapping.

**3. Dissected contributions of model components**

- Implemented transfer learning and frozen recurrent-weight experiments. We found that variability of working memory is rooted in recurrent dynamics, whereas bias can be flexibly shaped by readout weights. This dissociation provides novel insights on the neural mechanisms of working memory.

**4. Further clarified the results and modeling framework**

- Planned more statistical results and visualizations to better clarify the effects of reward in working memory resources.
- Planned to complete symbol definitions and extend the modeling explanation. A key contribution: Even after we extended the normative resource allocation model with a time domain (Sec. 3.3), the optimal allocation depends only on orientation—indicating a fixed strategy throughout the delay. Consequently, differences in resources for frequent (cardinal) vs. infrequent (diagonal) orientations accumulate over time, producing a stronger oblique effect with longer delays, consistent with our data.

We are pleased that all reviewers felt their concerns were addressed and most raised their scores, reflecting strengthened confidence in the work. We believe the planned revisions will improve clarity, rigor, and accessibility, and we thank the reviewers once again for helping shape a stronger paper.

---

### Decision · Program_Chairs · 2025-09-17

**Decision:**

Accept (poster)

**Comment:**

(a) Scientific Claims and Findings:
The paper presents a study combining behavioral data, a normative model, and RNN simulations to examine the interaction between RL and WM. Several key patterns of Human behaviours are reproduced by RNN models. The overall motivation is clear, and the authors provide useful clarifications, but some details still need attention for full clarity and reproducibility.

(b) Strengths of the Paper:

- The approach combines multiple angles, including behavioral data, modeling, and simulations, which strengthens the argument.
- The motivation behind the study is solid and well-articulated.

(c) Weaknesses and Missing Aspects:

- In some figures (like Fig. 2), differences between baseline and reward contexts aren't visually obvious, and statistical comparisons are missing.
- Some key symbols (like epsilon, kappa, etc.) aren’t defined, making it harder for others to reproduce the work.
- The explicit formula for J_opt is unclear, and the paper doesn't show whether relaxing the L2 penalty in the RNN would change the results.
- The sample size (14 participants) is small, which limits generalizability.

(d) Rebuttal Discussion and Changes:
The authors addressed most of the concerns raised, which led to a positive shift in scores. However, due to the rebuttal format, it's hard to assess how the revisions would impact the paper's readability and presentation fully. A more concrete update would help reviewers evaluate the changes better.

(e) Reasons for Acceptance/Rejection:

I recommand **accepting** this paper. The paper addresses an interesting and well-motivated problem. The novelty lies at the integration of human psychophysical experiments and ideal observer modeling. While there are still a few issues with clarity, reproducibility, and generalizability, the authors have made significant progress in addressing reviewer concerns.